# Interstellar formation of lactaldehyde, a key intermediate in the methylglyoxal pathway

Jia Wang [1,2], Chaojiang Zhang[1,2], Joshua H. Marks[1,2], Mikhail M. Evseev[3], Oleg V. Kuznetsov[3], Ivan O. Antonov[3] ✉ & Ralf I. Kaiser [1,2] ✉

Aldehydes are ubiquitous in star-forming regions and carbonaceous chondrites, serving as essential intermediates in metabolic pathways and molecular mass growth processes to vital biomolecules necessary for the origins of life. However, their interstellar formation mechanisms have remained largely elusive. Here, we unveil the formation of lactaldehyde ($CH_3CH(OH)CHO$) by barrierless recombination of formyl ($H\dot{C}O$) and 1-hydroxyethyl ($CH_3\dot{C}HOH$) radicals in interstellar ice analogs composed of carbon monoxide (CO) and ethanol ($CH_3CH_2OH$). Lactaldehyde and its isomers 3-hydroxypropanal ($HOCH_2CH_2CHO$), ethyl formate ($CH_3CH_2OCHO$), and 1,3-propenediol ($HOCH_2CHCHOH$) are identified in the gas phase utilizing isomer-selective photoionization reflectron time-of-flight mass spectrometry and isotopic substitution studies. These findings reveal fundamental formation pathways for complex, biologically relevant aldehydes through non-equilibrium reactions in interstellar environments. Once synthesized, lactaldehyde can act as a key precursor to critical biomolecules such as sugars, sugar acids, and amino acids in deep space.

Since the identification of the simplest aldehyde—formaldehyde ($H_2CO$, **1**)—in the interstellar medium (ISM) by Snyder et al. more than half a century ago (1969)[1], aldehydes (RCHO), with R being organic groups, have received extensive attention from the astronomy[2–4], astrobiology[5,6], astrochemistry[7–9], and physical organic chemistry communities[10–12] due to their role as crucial intermediates in metabolic pathways[13] and in molecular mass growth processes to vital biomolecules necessary for the origins of life[8,14,15]. Although the deep ocean hydrothermal vents are considered a likely scenario for the emergence of life[16–18], a substantial fraction of the prebiotic organic molecules on proto-Earth may have been of extraterrestrial origin[19]. In prebiotic chemistry, **1** can be converted to glycolaldehyde ($HOCH_2CHO$, **2**) in the formose or Butlerov reaction[14,20] thus serving as a starting material for the synthesis of complex sugars[21,22]. Mediated via quantum tunneling, acetaldehyde ($CH_3CHO$, **3**) can react with methanol ($CH_3OH$, **4**) to form the hemiacetal 1-methoxyethanol ($CH_3OCH(OH)CH_3$), a precursor to sugars and sugar-related molecules[11]. Triggered by energetic radiation, **3** reacts with carbon dioxide ($CO_2$) to form biorelevant

pyruvic acid ($CH_3COCOOH$, **5**), which is a vital molecule for metabolism processes in modern biochemistry[23]. Besides the nine aldehydes identified in the ISM (Fig. 1)[24], sixteen aldehydes including **1**, **3**, and propionaldehyde ($CH_3CH_2CHO$, **6**) have been detected in carbonaceous chondrites (Fig. 1)[9]. This indicates that aldehydes can not only be synthesized in deep space, but also survive the entrance of the parent bodies of the meteorites into the atmosphere of the early Earth[25]; this process could have provided an exogenous source of prebiotic molecules for the early evolution of life. Although aldehydes are of particular significance to the synthesis of astrobiologically relevant molecules, the fundamental formation routes of these molecules in the interstellar environment have remained largely elusive, especially the abiotic synthesis of the biomolecule lactaldehyde ($CH_3CH(OH)CHO$, **7**).

In prebiotic chemistry, **7** can form methylglyoxal ($CH_3COCHO$, **8**) through oxidation and further yield **5**—a molecular building block of metabolites and amino acids (Fig. 2). Through nucleophilic addition, **7** reacts with hydrogen cyanide (HCN) and ammonia ($NH_3$) to produce

[1]W. M. Keck Research Laboratory in Astrochemistry, University of Hawaii at Manoa, Honolulu, HI, USA. [2]Department of Chemistry, University of Hawaii at Manoa, Honolulu, HI, USA. [3]Samara National Research University, Samara, Russia. ✉e-mail: pfizeke@gmail.com; ralfk@hawaii.edu

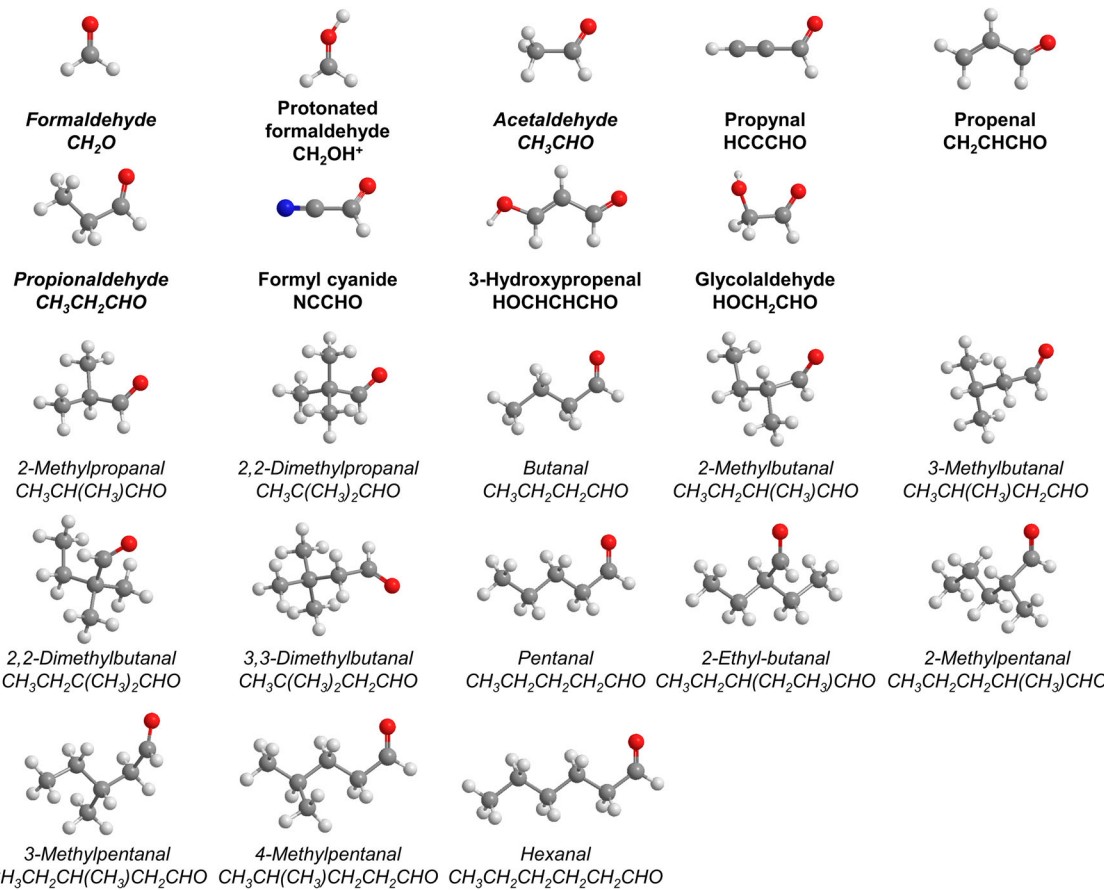

**Fig. 1 | Aldehydes identified in the interstellar medium (bold) and carbonaceous chondrites (italics).** The atoms are color-coded in white (hydrogen), gray (carbon), red (oxygen), and blue (nitrogen).

2,3-dihydroxybutanenitrile (CH₃CH(OH)CH(OH)CN, **9**) and 1-amino-1,2-propanediol (CH₃CH(OH)CH(OH)NH₂, **10**), respectively, which are molecular precursors to the proteinogenic amino acid threonine (NH₂CH(CH(OH)CH₃)COOH). Oxidation of **7** results in the formation of biomolecule lactic acid (CH₃CH(OH)COOH, **11**)[26], eventually contributing to the formation of sugar acids such as glyceric acid (HOCH₂CH(OH)COOH, **12**)[27]. Molecule **7** can be reduced to produce 1,2-propanediol (CH₃CH(OH)CH₂OH, **13**). Undergoing carbon–oxygen bond cleavage, **7** can be converted into **6**, which can react with two **1** molecules to form trimethylolethane (CH₃C(CH₂OH)₃) through condensation reactions[28]. Further, the cleavage of the carbon–carbon bond in **7** prepares **2**, which forms the simplest sugar molecule glyceraldehyde (HOCH₂CH(OH)CHO, **14**)[12] via aldol condensation. Consequently, **7** serves as a fundamental precursor to important biomolecules such as amino acids, sugars, and sugar acids (Fig. 2), which could have seeded the emergence of life on early Earth[29]. An elucidation of the interstellar formation of **7** is therefore of crucial importance to unraveling the synthesis routes of astrobiologically relevant molecules in deep space.

In this work, we demonstrate the bottom-up formation of **7** in interstellar ice analogs upon exposure to energetic irradiation in the form of proxies of galactic cosmic rays (GCRs). This is accomplished in low-temperature (5 K) carbon monoxide (CO)–ethanol (CH₃CH₂OH, **15**) ice mixtures through the barrierless radical–radical recombination of formyl (HĊO, **16**) with 1-hydroxyethyl (CH₃ĊHOH, **17**) radicals (Figs. 2 and 3). Combining VUV photoionization reflectron time-of-flight mass spectrometry (PI-ReToF-MS) and isotopic substitution studies, **7** along with its isomers 3-hydroxypropanal (HOCH₂CH₂CHO, **18**), ethyl formate (CH₃CH₂OCHO, **19**), and 1,3-propenediol (HOCH₂CHCHOH, **20**) were identified in the gas phase

during the temperature-programmed desorption (TPD) of the irradiated ice mixtures. As one of the most commonly detected molecules in interstellar ices, carbon monoxide was found with a fractional abundance of up to 55% with respect to water toward IRAS 08375−4109[30]. Ethanol (**15**) is abundant in the ISM and has been tentatively identified in the ices towards background stars[31] and young protostars with an abundance of up to 1.8% with respect to water[32]. Therefore, **7** and its isomers **18–20** form in interstellar ice composed of carbon monoxide and **15**. Once synthesized, these organics can be incorporated into planetesimals and may have eventually been delivered to planets such as early Earth through meteoritic impacts[33], providing an exogenous source for the formation of essential biorelevant molecules and thus helping to decipher the enigma of the molecular origins of life.

## Results

### Infrared spectroscopy

Fourier transform infrared (FTIR) spectroscopy was utilized to monitor the CO–CH₃CH₂OH ice mixture and its isotopically labeled system (CO–CD₃CD₂OD) before and after low dose (23 nA, 5 min) and high dose (123 nA, 10 min) irradiation at 5 K (Supplementary Figs. 1–4). Detailed assignments of FTIR spectra are compiled in Supplementary Tables 1–4. Prior to the electron irradiation, the absorptions are attributed to the fundamentals and combination modes of the reactants; the prominent absorptions in unprocessed CO–CH₃CH₂OH ice include the CO stretching (CO, 2136 cm⁻¹; ¹³CO, 2090 cm⁻¹) and overtone (4249 cm⁻¹) modes for carbon monoxide[34,35], and the broad O–H stretching mode (3050–3550 cm⁻¹), asymmetric C–H stretching mode (2977 cm⁻¹), symmetric C–H stretching modes (2935 cm⁻¹, 2900 cm⁻¹, and 2876 cm⁻¹), and the C–O stretching mode (1052 cm⁻¹) for ethanol[36].

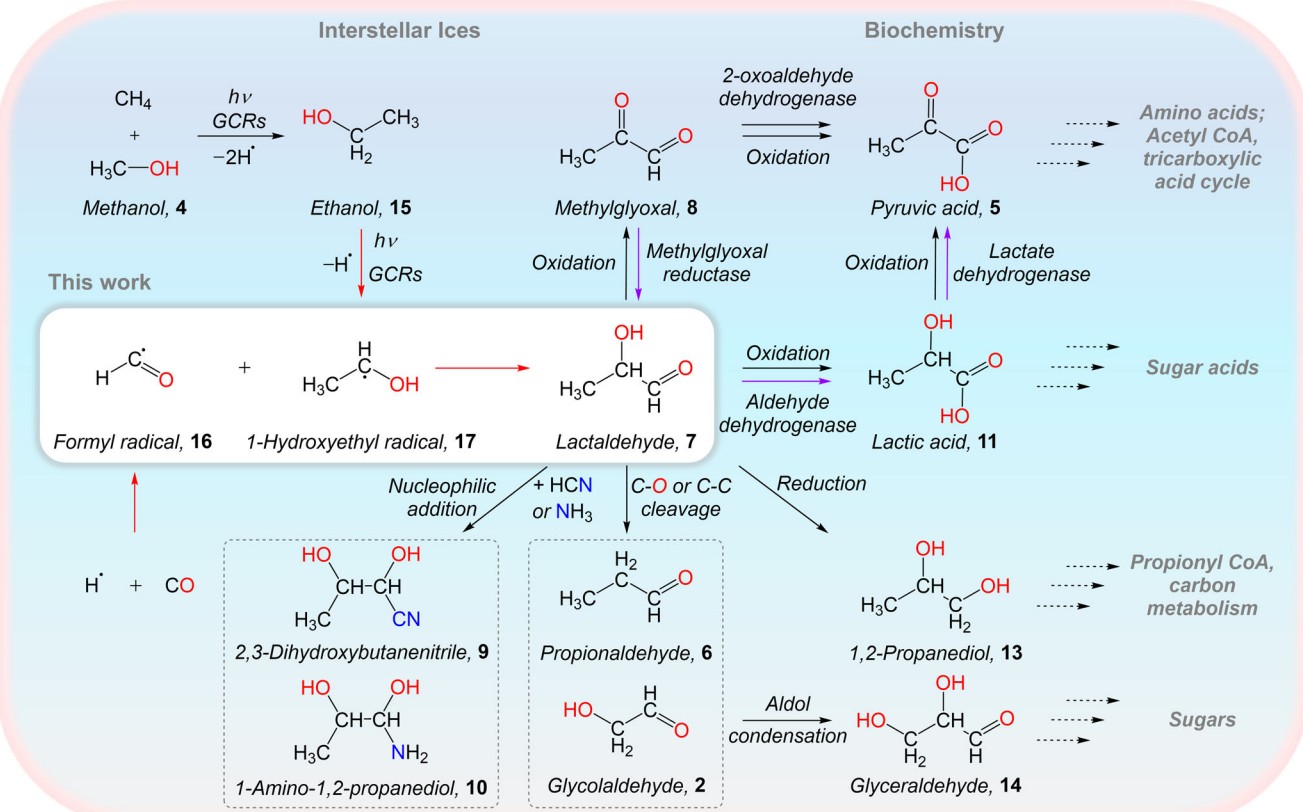

**Fig. 2 | Formation of lactaldehyde in interstellar ices and its role as a molecular building block of biorelevant molecules in the methylglyoxal cycle.** Lactaldehyde (**7**) is prepared in interstellar ice analogs composed of carbon monoxide and ethanol (**15**) through energetic processing by galactic cosmic ray proxies. This process involves carbon–carbon bond coupling via radical–radical recombination of the formyl (HĊO, **16**) with the 1-hydroxyethyl (CH₃ĊHOH, **17**) radical.

Lactaldehyde (**7**) serves as a precursor to sugar-related molecules such as glycolaldehyde (**2**) and lactic acid (**11**) thus contributing to the synthesis of sugars and sugar acids, respectively. In contemporary biochemistry, seven is a key intermediate in the methylglyoxal pathway (purple arrows) and a precursor to the formation of critical biorelevant molecules including pyruvic acid (**5**), methylglyoxal (**8**), and lactic acid (**11**), linking to the tricarboxylic acid (TCA) cycle.

After the irradiation, several absorption features emerged for the CO–CH₃CH₂OH ice mixtures (Supplementary Figs. 1–2). The absorption band at 2342 cm⁻¹ is assigned to the C=O stretching of carbon dioxide ($v_3$)[34]. The absorptions at 1853 cm⁻¹, 1843 cm⁻¹, 1726 cm⁻¹, and 1713 cm⁻¹ are linked to **16** (HĊO, $v_3$), *trans*-hydroxycarbonyl (HOĊO, $v_2$), **1** (H₂CO, $v_2$), and **3** (CH₃CHO, $v_4$), respectively[34,37,38]; these absorptions are shifted to 1796 cm⁻¹ for **16**-d₁ (DĊO, $v_3$), 1780 cm⁻¹ for *trans*-hydroxycarbonyl-d₁ (DOĊO, $v_2$), 1695 cm⁻¹ for **1**-d₂ (D₂CO, $v_2$), and 1715 cm⁻¹ for **3**-d₄ (CD₃CDO, $v_4$) in irradiated CO–CD₃CD₂OD ice mixtures (Supplementary Figs. 3 and 4)[10,34,37]. The assignments to these absorptions were further confirmed in irradiated ¹³CO–¹³CH¹³CH₂OH ice and C¹⁸O–CHCH₂OH ice (Supplementary Figs. 5 and 6 and Supplementary Tables 5 and 6)[34,37,39–41]. It is worth noting that the absorptions at 1431 cm⁻¹ and 1352 cm⁻¹ observed in high-dose irradiated CO–CH₃CH₂OH ice (Supplementary Fig. 2) can be tentatively associated with the 1-hydroxyethyl (**17**) radical based on calculated frequencies at 1426 cm⁻¹ ($v_7$) and 1344 cm⁻¹ ($v_9$) of **17** at the CCSD(T)/cc-pVTZ level of theory[42]. Due to the limited molecular mobility at 5 K, these radicals are preserved within the ice; radicals may not be able to react if they form without nearby radicals[43]. The absorptions at 1681 cm⁻¹ and 1580 cm⁻¹ in irradiated CO–CD₃CD₂OD ices may link to one or more carbonyl (C=O) containing species such as **7**, **18**, and **19**. Due to the overlapping absorption features of the formed complex organics during radiation processing, FTIR spectra cannot uniquely detect complex compounds such as **7** and its isomers alone, highlighting that an alternative experimental technique is needed to identify individual reaction products.

## Mass spectrometry

The photoionization reflectron time-of-flight mass spectrometry (PI-ReToF-MS) technique is exploited here to identify the reaction products isomer-specifically based on their desorption temperatures and adiabatic ionization energies (IEs)[10,44]. The PI-ReToF mass spectra of the irradiated carbon monoxide–ethanol (**15**) ices during TPD are compiled in Figs. 4 to 6 and Supplementary Fig. 7. Focusing on C₃H₆O₂ isomers, four photon energies at 11.10 eV, 10.23 eV, 9.71 eV, and 9.29 eV were selected to distinguish isomers **7**, **18**, and **19** formed via radical–radical recombination after the low dose irradiation (Fig. 3). At 11.10 eV, the TPD profile of ions at $m/z$ = 74 for CO–CH₃CH₂OH ice was deconvoluted by fitting to four split Pearson VII distributions peaking at 118 K, 144 K, 178 K, and 222 K, respectively (Fig. 4b). Given the molecular weights of the reactants in CO–CH₃CH₂OH ice, the ion signal at mass-to-charge ($m/z$) of 74 can belong to organic compounds with formulae including C₆H₂, C₄H₁₀O, C₃H₆O₂, and/or C₂H₂O₃. Hence it is imperative to confirm the molecular formula using isotopically labeled precursors. Utilizing the fully deuterated ices with CO–CD₃CD₂OD ice, the TPD profile of $m/z$ = 80 in irradiated CO–CD₃CD₂OD ice matches well with that of $m/z$ = 74 in irradiated CO–CH₃CH₂OH ice (Fig. 5a), indicating the presence of exactly six hydrogen atoms. In addition, the TPD profile of $m/z$ = 74 in irradiated CO–CH₃CH₂OH ice was found to undergo an isotopic mass shift to $m/z$ = 77 in fully carbon-13 isotopically labeled ice (¹³CO–¹³CH₃¹³CH₂OH), confirming the inclusion of exactly three carbon atoms. Furthermore, the TPD profile of $m/z$ = 74 in CO–CH₃CH₂OH ice shifts two atomic mass units (amu) to $m/z$ = 76 in C¹⁸O–CH₃CH₂OH ice (Supplementary Fig. 8), indicating the presence

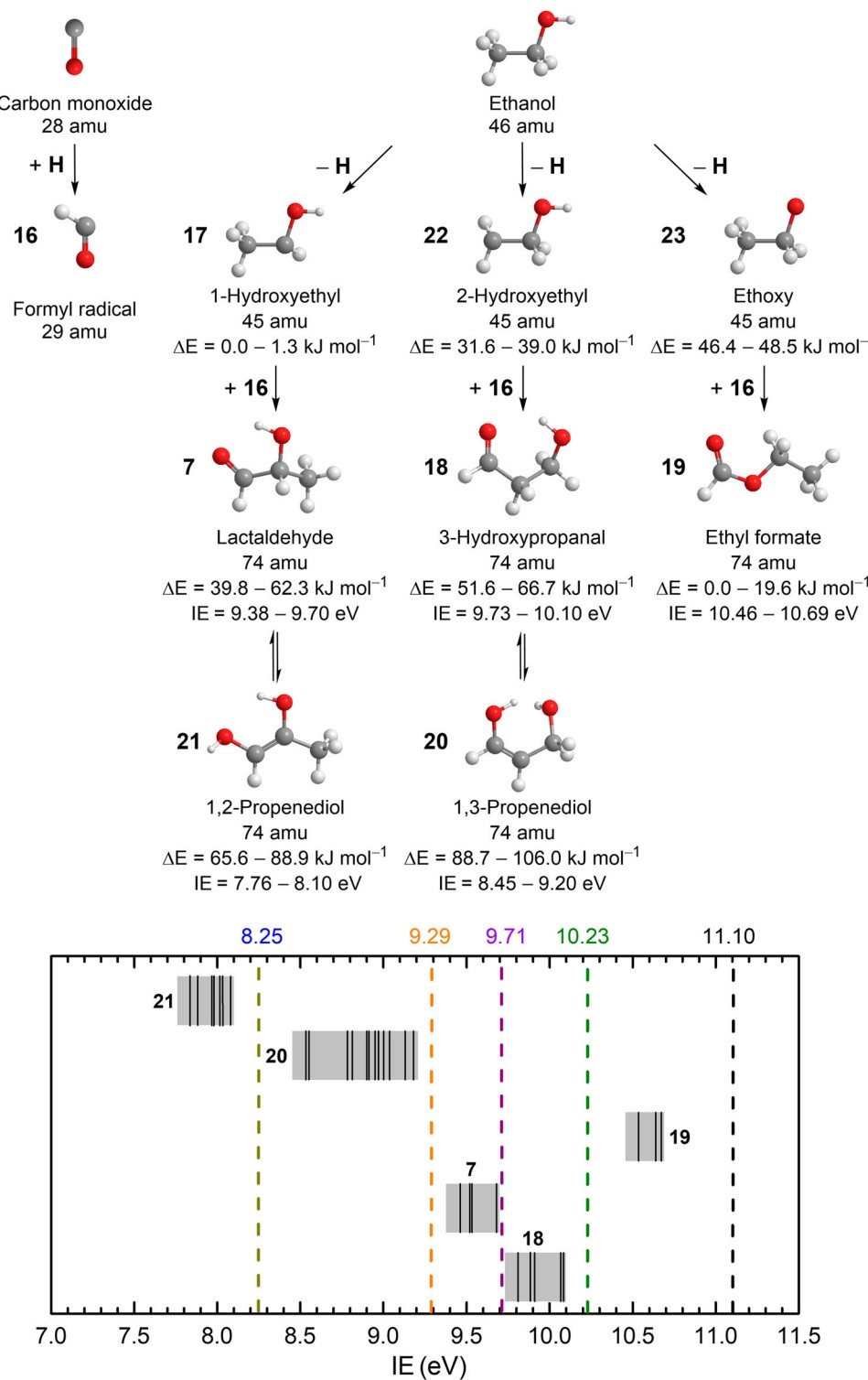

**Fig. 3 | Reaction scheme leading to five C₃H₆O₂ (*m/z* = 74) isomers in irradiated carbon monoxide–ethanol ices.** Barrierless radical–radical reactions of **16** with **17**, **22**, and **23** produce **7**, **18**, and **19**, respectively; tautomerization of **7** and **18** may lead to the enols **21** and **20** (top). The computed relative energies (Δ*E*) of radicals[47] and products are shown as ranges containing all conformers. The bottom figure compiles the computed adiabatic ionization energies (IEs) of isomers (black solid line) and ranges of their conformers (gray area) after error analysis (Supplementary Tables 7–12). Five VUV photon energies (dashed lines) were used to photoionize sublimed molecules during TPD.

of at least one oxygen atom. These findings validate the assignment of the reaction products of the molecular formula C₃H₆O₂ for the TPD profile (Supplementary Note 1).

As previously mentioned, the TPD profile of *m/z* = 74 (C₃H₆O₂⁺) at 11.10 eV in the low dose irradiated CO−CH₃CH₂OH ice (Fig. 4b) reveals peaks at 118 K (Peak I), 144 K (Peak II), 178 K (Peak III), and 222 K (Peak IV). Note that the peak sublimation temperatures of acetaldehyde (**3**) at *m/z* = 44 and ethanol (**15**) at *m/z* = 46 are at 118 K[10] and 149 K, respectively (Supplementary Fig. 9), indicating that Peaks I and II are due to the co-sublimation of acetaldehyde (**3**) and **15**, respectively. A

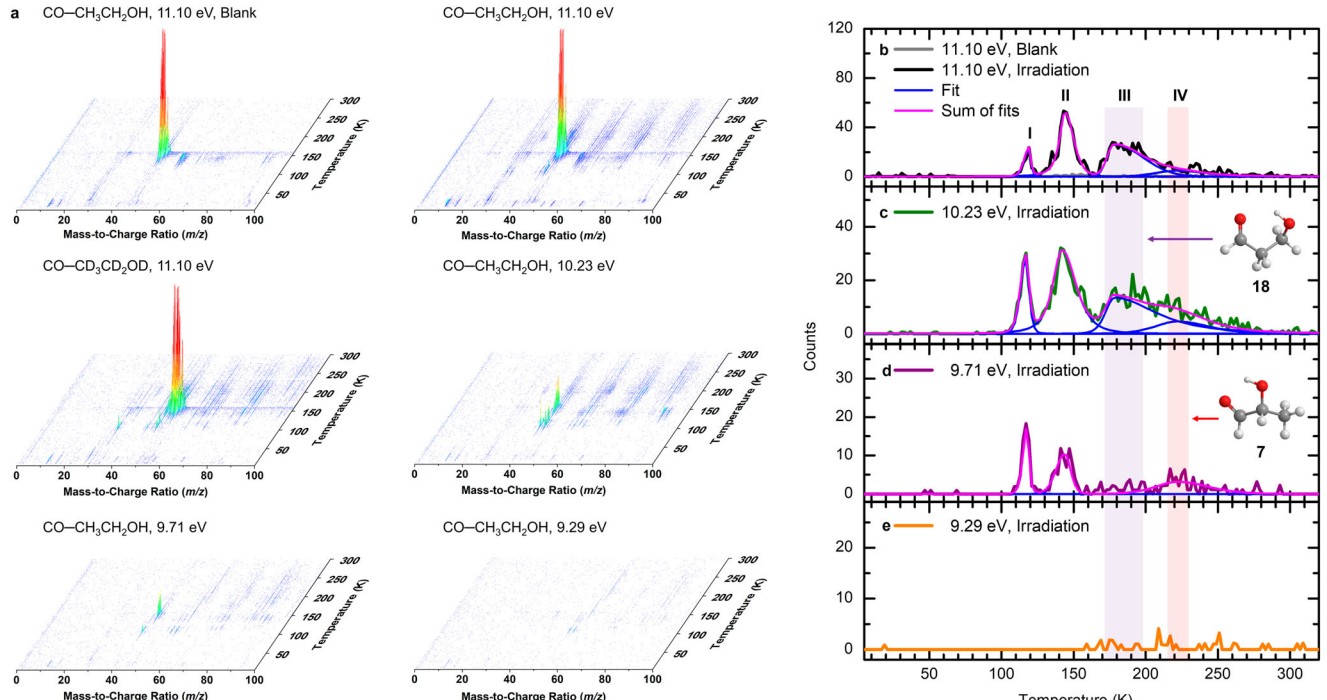

**Fig. 4 | PI-ReToF-MS data during TPD of carbon monoxide–ethanol ices with low dose (23 nA, 5 min) irradiation.** Data were recorded for the unirradiated (blank) CO–CH₃CH₂OH ice at 11.10 eV, the irradiated CO–CH₃CH₂OH ice at 11.10 eV, 10.23 eV, 9.71 eV, and 9.29 eV, and the irradiated CO–CD₃CD₂OD ice at 11.10 eV (**a**). TPD profiles of m/z = 74 in CO–CH₃CH₂OH ice were measured at 11.10 eV (**b**), 10.23 eV (**c**), 9.71 eV (**d**), and 9.29 eV (**e**). The solid magenta lines indicate the total fit of the spectra. At 11.10 eV, the TPD profile can fit with four Peaks I–IV. The red and purple shaded regions indicate the peak positions corresponding to lactaldehyde (**7**) and 3-hydroxypropanal (**18**), respectively.

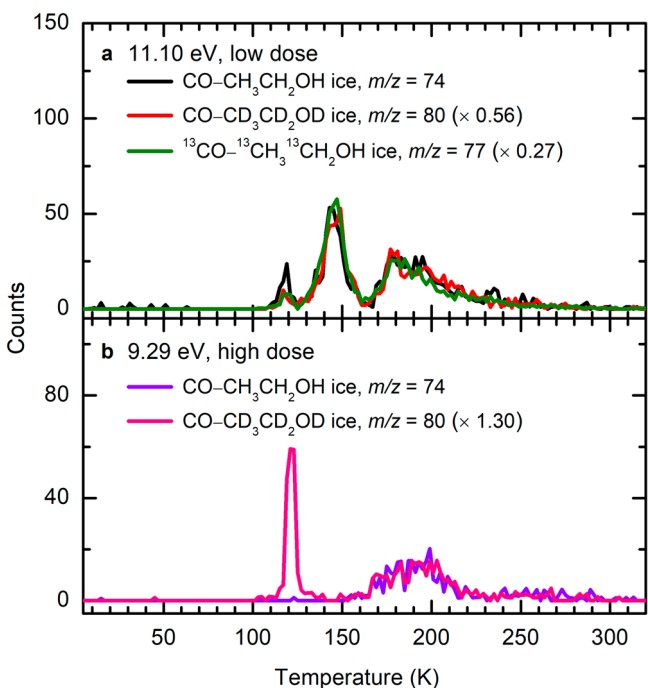

**Fig. 5 | Ion signal during TPD of isotopically labeled carbon monoxide–ethanol ices as a function of temperature.** TPD profiles were measured at 11.10 eV with low dose irradiation (**a**) and at 9.29 eV with high dose irradiation (**b**).

blank experiment without electron irradiation of the ice was carried out under identical conditions (Fig. 4b); no sublimation event was detected, confirming that Peaks I–IV is caused by the electron irradiation of the ice. At 11.10 eV, all isomers **7** (IE = 9.38–9.70 eV), **18**

(IE = 9.73–10.10 eV), **19** (IE = 10.46–10.69 eV), **20** (IE = 8.45–9.20 eV), and 1,2-propenediol (HOCHCH(OH)CH₃, **21**; IE = 7.76–8.10 eV) can be photoionized (Fig. 3 and Supplementary Tables 7–12). Therefore, these peaks can be associated with any isomers **7, 18–21**. Thereafter, the photon energy was reduced to 10.23 eV, at which isomer **19** (IE = 10.46–10.69 eV) cannot be ionized. At 10.23 eV, ion signals of Peaks I–IV remain (Fig. 4c), indicating that **19** is not present in detectable quantities and hence not formed. Upon reducing the photon energy further to 9.71 eV, at which isomers **7** (IE = 9.38–9.70 eV), **20** (IE = 8.45–9.20 eV), and **21** (IE = 7.76–8.10 eV) can be ionized but not isomer **18** (IE = 9.73–10.10 eV), Peak III at 178 K vanishes (Fig. 4d). Therefore, Peak III can be associated with **18**. Further lowering the photon energy to 9.29 eV, at which enols **20** and **21** can be ionized but not isomer **7**, no sublimation events were observed (Fig. 4e), suggesting that the ion signals of Peaks I, II, and IV recorded at 9.71 eV can be linked to **7**; no evidence for **20** and **21** can be provided in low dose irradiation experiments. In summary, the low-dose irradiation experiments revealed the formation of isomers **7** and **18**.

To further probe the synthesis of isomers **19–21**, high-dose experiments were carried out (Fig. 6). Compared with the results in low-dose irradiated CO–CH₃CH₂OH ice at 11.10 eV, the TPD profile of m/z = 74 shows a sublimation event peaking at 128 K (Peak V) (Fig. 6b). To assist in the identification of isomer **19**, a calibration experiment without irradiation (blank) was performed at 11.10 eV by adding 1% of **19** (IE = 10.53–10.63 eV)[45] into the reactants under identical experimental conditions. The TPD profile of **19** at m/z = 74 shows two sublimation events peaking at 131 K and 149 K (Fig. 6c); the latter peak is caused by the co-sublimation with ethanol (**15**). The first sublimation event (131 K) of **19** matches Peak V (128 K), indicating that Peak V can be associated with **19**. Upon reducing the photon energy to 9.29 eV, the TPD profile of m/z = 74 (Fig. 6d) shows a sublimation event that starts at 160 K, peaks at 197 K, and returns to the baseline level at 220 K (Peak VI). Recall that no sublimation event was observed at 9.29 eV in low

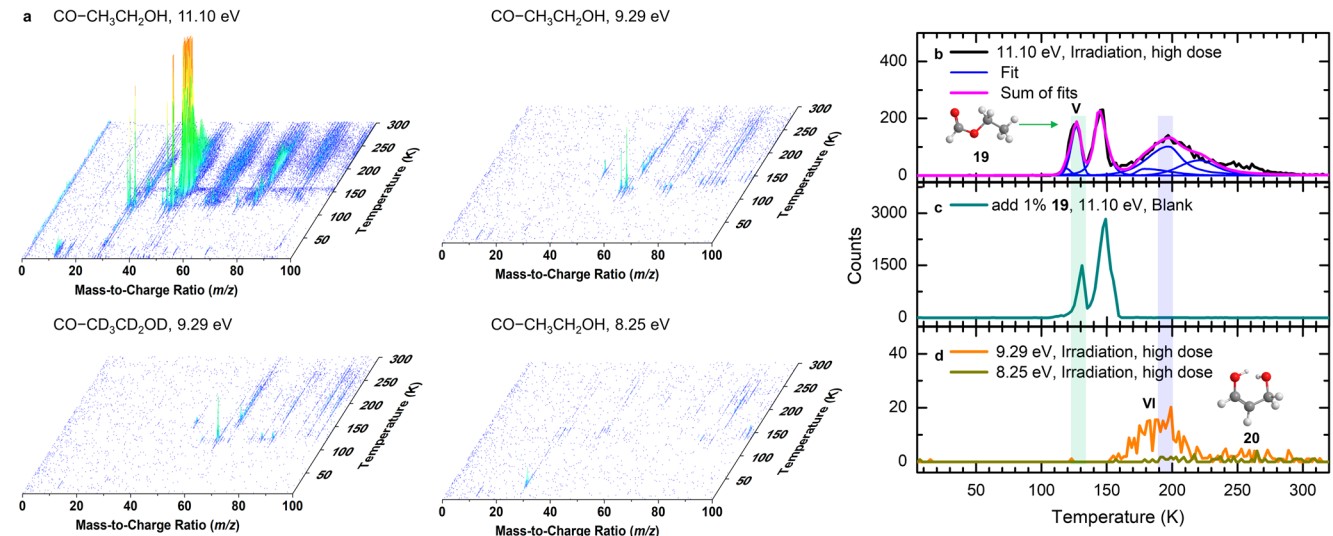

**Fig. 6 | PI-ReToF-MS data during the TPD of carbon monoxide–ethanol ices with higher dose (123 nA, 10 min) irradiation.** Data were recorded for the irradiated CO−CH₃CH₂OH ice at 11.10 eV, 9.29 eV, and 8.25 eV, and the irradiated CO−CD₃CD₂OD ice at 9.29 eV (**a**). TPD profiles of $m/z = 74$ in CO−CH₃CH₂OH ice were measured at 11.10 eV (**b**), 9.29 eV, and 8.25 eV (**d**). TPD profile of $m/z = 74$ in a blank experiment with 1% ethyl formate (**19**) was recorded at 11.10 eV (**c**). The solid magenta line indicates the total fit of the spectra. The green and blue shaded regions indicate the peak positions corresponding to **19** and 1,3-propenediol (**20**), respectively.

dose experiment, it is necessary to confirm the formula of this ion signal using isotopically labeled ice. The substitution of CH₃CH₂OH by CD₃CD₂OD results in products with six deuterium atoms that can be observed at $m/z = 80$ in the CO−CD₃CD₂OD ice (Fig. 5b). The first event peaking at 121 K in the TPD profile of $m/z = 80$ is linked to C₄H₈O isomers (Supplementary Fig. 10). By matching the TPD profiles for the deuterated molecules in irradiated CO−CD₃CD₂OD ice (C₃D₆O₂⁺, $m/z = 80$), the assignment of Peak VI can be clearly connected with C₃H₆O₂ isomers. Since only enols **20** (IE = 8.45–9.20 eV) and **21** (IE = 7.76–8.10 eV) can be ionized at 9.29 eV, Peak VI can be linked to **20** and/or **21**. To further identify **20** and **21**, we lowered the photon energy to 8.25 eV, at which **20** cannot be ionized. In comparison to the result at 9.29 eV, Peak VI vanishes at 8.25 eV (Fig. 6d), suggesting that this sublimation event peaking at 197 K must be linked to **20**; no evidence for **21** could be provided. Overall, the low-dose irradiation experiments revealed the formation of isomers **7** and **18**, whereas the high-dose studies further revealed the formation of **19** and enol **20**.

## Discussion

Having provided compelling evidence on the formation of isomers **7** and **18**–**20**, we now focus on their potential formation mechanisms. First, upon interaction with GCRs, the unimolecular decomposition of ethanol (**15**) can lead to the formation of atomic hydrogen (Ḣ) plus 1-hydroxyethyl (**17**), 2-hydroxyethyl (**22**), and/or ethoxy (**23**) radicals[46,47]. Reactions [1–3] are endoergic by 390 kJ mol⁻¹ to 437 kJ mol⁻¹ [48,49] compensated by the energy from the impinging electrons. Recall that **17** was tentatively assigned via the absorptions at 1431 cm⁻¹ ($v_7$) and 1352 cm⁻¹ ($v_9$) in the irradiated CO−CH₃CH₂OH ice. Reactions [1–3] closely resemble the decomposition of methanol (CH₃OH) forming the hydroxymethyl (ĊH₂OH) and methoxy (CH₃Ȯ) radicals via hydrogen–carbon and –oxygen bond cleavages[10,50]. The hydrogen atoms can then be added to the carbon monoxide, forming the formyl (**16**) via reaction [4] with a reaction exoergicity of 61 kJ mol⁻¹. Previous work by Bennett et al. suggested an entrance barrier for the reaction [4] to be 11 kJ mol⁻¹ (0.114 eV)[51]. This entrance barrier can be overcome by the suprathermal hydrogen atoms having excess kinetic energies of a few eV leading to the formation of **16** as identified via FTIR spectroscopy at 1853 cm⁻¹ (HĊO, $v_3$)[37,38] in

irradiated CO−CH₃CH₂OH ice and at 1796 cm⁻¹ for **16**-d₁ (DĊO, $v_3$) in irradiated CO−CD₃CD₂OD ice[37].

$$CH_3CH_2OH(\mathbf{15}) \rightarrow CH_3\dot{C}HOH(\mathbf{17}) + \dot{H} \quad (+390\ \mathrm{kJ\,mol}^{-1}) \quad (1)$$

$$CH_3CH_2OH(\mathbf{15}) \rightarrow \dot{C}H_2CH_2OH(\mathbf{22}) + \dot{H} \quad (+422\ \mathrm{kJ\,mol}^{-1}) \quad (2)$$

$$CH_3CH_2OH(\mathbf{15}) \rightarrow CH_3CH_2\dot{O}(\mathbf{23}) + \dot{H} \quad (+437\ \mathrm{kJ\,mol}^{-1}) \quad (3)$$

$$\dot{H} + CO \rightarrow H\dot{C}O(\mathbf{16}) \quad (-61\ \mathrm{kJ\,mol}^{-1}) \quad (4)$$

Second, **7**, **18**, and **19** form via barrierless radical–radical recombination via reactions [5–7]. These reactions are exoergic by 334 kJ mol⁻¹ to 420 kJ mol⁻¹ [48,49]. Due to the limited molecular mobility caused by the low temperatures of 5 K, these radicals are preserved within the ice; radicals may not be able to react if they form without nearby radicals[43]. This is especially relevant for relatively large radicals such as **17**, **22**, and **23**, indicating that the formation of lactaldehyde and its isomers is likely to proceed via non-diffusive radical recombination mechanisms[52]. However, at higher temperatures such as 20 K, the diffusive mechanism may compete with the non-diffusive pathways as radicals such as HĊO become more mobile[52,53]. Recall that the TPD profile of $m/z = 74$ (C₃H₆O₂⁺) in CO−CH₃CH₂OH ice shifts 2 amu to $m/z = 76$ (C₃H₆O¹⁸O⁺) in C¹⁸O−CH₃CH₂OH ice, indicating one carbon monoxide and one **15** molecules are involved in the formation of the products. Recent studies by Zasimov et al. revealed the predominant formation of 1-hydroxyethyl (**17**) resulting from the X-ray irradiation of matrix-isolated **15** molecules[43,44]. Although indirect evidence was provided for the primary formation of ethoxy (**23**), the isomerization of **23** to form **17** is rapid even at 7 K due to tunneling[46]. This agrees with our experimental results suggesting that **19** resulting from the recombination of **16** with **23** remained undetectable at low dose studies and

can only be detected in high-dose irradiation experiments.

$$H\dot{C}O(16) + CH_3\dot{C}HOH(17) \rightarrow CH_3CH(OH)CHO\,(7) \quad (-334\,kJ\,mol^{-1})$$
(5)

$$H\dot{C}O(16) + \dot{C}H_2CH_2OH(22) \rightarrow HOCH_2CH_2CHO\,(18) \quad (-354\,kJ\,mol^{-1})$$
(6)

$$H\dot{C}O(16) + CH_3CH_2\dot{O}(23) \rightarrow CH_3CH_2OCHO\,(19) \quad (-420\,kJ\,mol^{-1})$$
(7)

Third, through hydrogen migrations, **18** can tautomerize to enol **20** with a reaction endoergicity of 37 kJ mol⁻¹ via reaction [8][48,49]. The barrier of keto-enol tautomerization can be overcome by the energy contributed by GCR proxies[35]. Although **7** may tautomerize to **21**, no evidence of **21** was observed in our experiments. Previous studies revealed that the interconversion of **7** can bypass the enol **21** and access its more stable ketone isomer hydroxyacetone (CH₃C(O)CH₂OH, **24**)[54]. However, the identification of **24** is inconclusive under current experimental conditions (Supplementary Note 2).

$$HOCH_2CH_2CHO(18) \leftrightharpoons HOCH_2CHCHOH(20) \rightarrow \quad (-37\,kJ\,mol^{-1})$$
(8)

Altogether, we present the bottom-up formation pathways of lactaldehyde (**7**) and its isomers (**18–20**) in low-temperature carbon monoxide−ethanol ice mixtures upon exposure to energetic electrons, which simulate secondary electrons produced by GCRs as they penetrate ices within a cold molecular cloud aged up to seven million years[55]. These molecules were identified in the gas phase during the TPD phase utilizing photoionization reflectron time-of-flight mass spectrometry (PI-ReToF-MS) along with isotopic substitution experiments. The CO−CH₃CH₂OH ices selected in our laboratory simulations present model ice to understand the formation pathways of lactaldehyde and its isomers in a comprehensive way. Future experiments can explore the effects of ice composition on these molecules by incorporating other simple molecules common to interstellar ice such as water (H₂O), carbon dioxide (CO₂), and methanol (CH₃OH) into the ice mixture. Isomers **7, 18**, and **19** were formed via radical−radical recombination of the formyl (H$\dot{C}$O, **14**) with the 1-hydroxyethyl (**17**), 2-hydroxyethyl ($\dot{C}$H₂CH₂OH, **22**), and ethoxy (CH₃CH₂$\dot{O}$, **23**) radicals, respectively; enol **20** was accessed through keto-enol tautomerization of **18**. These findings provide fundamental steps toward the understanding of the fundamental formation mechanisms of complex aldehydes (RCHO) and their enols (RC = C(OH)H) under astrophysical conditions.

Within cold molecular clouds, galactic cosmic rays can trigger non-equilibrium reactions in the interstellar ice that are composed of simple molecules such as water (H₂O), carbon dioxide (CO₂), carbon monoxide (CO), methanol (CH₃OH), and ammonia (NH₃)[55,56]. Carbon monoxide and ethanol (**15**) are abundant in the interstellar environment[24]. In interstellar ices, carbon monoxide has a fractional abundance of up to 55% with respect to water[30], and **15** has been tentatively identified with an abundance of up to 1.8% with respect to water[32]. Therefore, through the feasible formation mechanisms as demonstrated here, **7** and its isomers **18–20** can form abiotically in interstellar ices containing carbon monoxide and **15** upon interaction with ionizing radiation. Once formed, these molecules can sublime into the gas phase in the hot core stage, representing promising candidates for future astronomical searches via radio telescopes such as the Atacama Large Millimeter Array (ALMA). In fact, isomer **18** was identified toward Sagittarius B2(N)[57] and Orion[58]−two star-forming regions. Laboratory simulation experiments by Marcellus et al. revealed the identification of **7** in the simulated pre-cometary organic residues after the VUV irradiation of an ice mixture containing water,

methanol, and ammonia[59]. Although **7** is one of the simplest chiral molecules that could reasonably exist in deep space, it has not yet been detected[24]. Recent work by Alonso et al. searched for **7** in three high-mass star-forming regions and reported an upper limit of fractional abundance of $1 \times 10^{-7}$ with respect to hydrogen (H₂)[4].

In biochemistry, lactaldehyde (**7**) serves as a key intermediate in the methylglyoxal pathway, in which glucose (C₆H₁₂O₆) can be converted into pyruvate[60]. The methylglyoxal pathway is strictly linked to glycolysis[61,62], a contemporary biochemical process vital to cellular metabolism. After being formed from methylglyoxal (**8**) via methylglyoxal reductase[63], **7** can be converted into lactic acid (**11**) through aldehyde dehydrogenase[64] and then leads to the formation of **5** via lactate dehydrogenase (Fig. 2)[63], which along with its deprotonated pyruvate anion (CH₃COCOO⁻) are critical molecules linked to the tricarboxylic acid (TCA) cycle[65]. In addition, **7** can be reduced to form 1,2-propanediol (**13**) and acts as an intermediate in the 1,2-propanediol synthesis pathway[66], contributing to the carbon metabolism[13]. Our laboratory experiments revealed that biorelevant aldehydes such as **7** can form through radical−radical recombination from carbon monoxide and alcohols on interstellar ice grains, serving as fundamental precursors to important biomolecules such as sugars, sugar acids, and amino acids. **7** could produce the sugar-related molecule **2**, which can be formed via the recombination of **16** and hydroxymethyl radicals within ices of **4** and carbon monoxide−**4** upon exposure to ionizing radiation[67,68]. Recent studies revealed that the thermal reaction of **3** with ammonia yields 1-aminoethanol (CH₃CH(OH)NH₂) at a low temperature of 65 K, contributing to the synthesis of prebiotic chelating agents[5]. Similarly, **7** could react with ammonia through a nucleophilic reaction to form 1-amino-1,2-propanediol (**12**), a precursor to the proteinogenic amino acid threonine. Once formed within cold molecular clouds, these molecules can be eventually incorporated into planetoids, asteroids, and comets[23,69], and ultimately delivered to planets like the early Earth, providing an exogenous source of prebiotic molecules[25]. In fact, extraterrestrial aldehydes, sugars, and amino acids have been detected in carbonaceous chondrites[9,33,70]. Under prebiotic conditions, the presence of **7** can contribute to the synthesis of biomolecules such as **5** and **9** (Fig. 2), thus critically advancing our fundamental knowledge of the synthesis routes to critical biorelevant molecules in deep space and on prebiotic Earth.

## Methods
### Experimental
All experiments were conducted in a stainless steel ultrahigh vacuum (UHV) chamber maintained at pressures of $5 \times 10^{-11}$ Torr by magnetically levitated turbomolecular pumps (Osaka, TG1300MUCWB, TG420MCAB) backed by a hydrocarbon-free dry scroll pump (XDS35i, BOC Edwards)[27]. A polished silver substrate (12.6 × 15.1 mm²) was interfaced to a cold head that was cooled to 5 K by a two-stage closed-cycle helium refrigerator (Sumitomo Heavy Industries, RDK-415E). The cold head can rotate freely and translate vertically through a doubly differentially pumped rotational feedthrough (Thermionics Vacuum Products, RNN-600/FA/ MCO) and an adjustable bellows (McAllister, BLT86). Ethanol (C₂H₅OH; Pharmco, anhydrous, ≥99.5% purity), ethanol-d₆ (C₂D₅OD; Cambridge Isotope Laboratories, anhydrous, 99% atom D), or ethanol-¹³C₂ (Sigma Aldrich, 99 atom% ¹³C) was filled into a glass vial interfaced to a UHV chamber. The samples were subjected to several freeze-thaw cycles to remove residual atmospheric gases using liquid nitrogen. Carbon monoxide (CO; Sigma Aldrich, >99%), carbon monoxide-¹³C (¹³CO; Sigma Aldrich, ≥99 atom% ¹³C, ≤6 atom% ¹⁸O), or carbon monoxide-¹⁸O (C¹⁸O; Sigma Aldrich, 99.9 atom% ¹²C, 95 atom% ¹⁸O) was premixed with ethanol, ethanol−d₆, or ethanol-¹³C₂ vapor to prepare a gas mixture with a 2:1 ratio of carbon monoxide to ethanol. To prepare the ice, the gas mixture was leaked into the main chamber at pressures of $4 \times 10^{-8}$ Torr via a glass capillary array and deposited onto the silver substrate, which was cooled to 5 K. Although the

temperatures of 5 K used in these experiments are slightly lower than that typically found in molecular clouds, intact reactive intermediates can be preserved to provide valuable mechanistic insights in such cold ice. Laser interferometry was used to monitor the ice thickness during the deposition via a photodiode and a helium-neon laser (632.8 nm)[71]. The average index of $1.26 \pm 0.04$ was used to derive the thickness of the mixture ice (CO–CH$_3$CH$_2$OH) from the refractive indexes of carbon monoxide ice ($n = 1.25 \pm 0.03$)[34] and that of ethanol ice ($n = 1.26$)[72]. This average index was used for the isotopically labeled ice mixtures. The ice thicknesses of CO–CH$_3$CH$_2$OH ice were determined to be $880 \pm 50$ nm. The densities of $0.80 \pm 0.01$ g cm$^{-3}$ for CO ice[34] and $0.584$ g cm$^{-3}$ for CH$_3$CH$_2$OH ice[72] were used. For isotopically labeled ice mixtures, variations in density were considered based on the masses of reactants. A Fourier transform infrared (FTIR) spectrometer (Thermo Electron, Nicolet 6700) measured the ice after deposition in the range of $6000 – 500$ cm$^{-1}$ with a resolution of $4$ cm$^{-1}$. FTIR spectra of pure ethanol, ethanol-d$_6$, and ethanol-$^{13}$C$_2$ ices were collected at 5 K with thicknesses of $760 \pm 50$ nm, $810 \pm 50$ nm, and $450 \pm 50$ nm, respectively (Supplementary Figs. 11–13), which were used to determine the thicknesses and column densities of ethanol, ethanol-d$_6$, and ethanol-$^{13}$C$_2$ in the mixture ices based on the integrated area of multiple absorption bands. Utilizing the integrated infrared absorptions of carbon monoxide at 2091 cm$^{-1}$ ($v_1$, $^{13}$CO, $1.32 \times 10^{-17}$ cm molecule$^{-1}$) and 4249 cm$^{-1}$ ($2v_1$, CO, $1.04 \times 10^{-19}$ cm molecule$^{-1}$)[34] and the absorption bands of pure ethanol ices with known thickness (Supplementary Figs. 11–13 and Supplementary Tables 13–15), the ratio of carbon monoxide to ethanol in the ice mixture (CO–CH$_3$CH$_2$OH) was estimated to be $2.5 \pm 0.4{:}1$ (Supplementary Table 16). It is necessary to note that the absorption coefficients of carbon monoxide were obtained via transmission absorption IR spectroscopy, which may differ from those obtained using reflection absorption IR spectroscopy[73,74]. Other factors, such as the thickness of the ice and the angle of incidence of the IR beam, can affect the relative peak heights in reflectance IR spectra. Here, we use the absorption coefficients in reflection as a means to estimate the ratio of components in the ice mixtures. Although the ratio of carbon monoxide to ethanol used in the experiments may not be a typical abundance ratio observed in molecule clouds, this ratio ensures the highest possible yield of C$_3$H$_6$O$_2$ isomers and thus facilitates their detection.

After deposition, the ice mixtures were exposed to 5 keV electrons released from an electron gun (SPECS, EQ PU-22); the electron beam was scanned over an area of 1.6 cm$^2$ for low dose (23 nA, 5 min) and high dose (123 nA, 10 min) irradiations. Prior to irradiation, a phosphor screen was used to monitor and adjust the electron beam, ensuring uniform exposure across the sample substrate. The electron beam current was measured using a Faraday cup before and after irradiation, with fluctuations kept within 3 nA. Based on Monte Carlo simulations carried out with the CASINO software suite[75], the high dose irradiation for CO–CH$_3$CH$_2$OH ice corresponds to doses of $1.50 \pm 0.25$ eV molecule$^{-1}$ for CO and $3.31 \pm 0.54$ molecule$^{-1}$ for CH$_3$CH$_2$OH, respectively (Supplementary Table 16). These doses simulate secondary electrons generated in the track of galactic cosmic rays (GCRs) in cold molecular clouds (Supplementary Note 3) aged up to $7 \times 10^6$ years[55]. The average penetration depth of electrons in CO–CH$_3$CH$_2$OH ice was calculated to be $420 \pm 70$ nm using CASINO 2.42[75]; 99% of the electron energy was deposited in the top $710 \pm 50$ nm sample layers, which is less than the ice thickness ($880 \pm 50$ nm), preventing the interaction between the substrate and electrons. The infrared spectra of ice were measured by the FTIR spectrometer in situ before, during, and after irradiation.

After irradiation, the ices were heated from 5 K to 320 K at 1 K min$^{-1}$ during the temperature-programmed desorption (TPD) process. Subliming molecules were photoionized by pulsed 30 Hz vacuum ultraviolet (VUV) lights, which were generated through resonant four-wave mixing schemes inside the noble gas jet[10]. The VUV photons were generated via sum frequency generation ($2\omega_1 + \omega_2$; 11.10 eV) and difference frequency generation ($2\omega_1 - \omega_2$; 10.23 eV, 9.71 eV, 9.29 eV, and 8.25 eV) utilizing two Nd:YAG lasers (Spectra-Physics, Quanta Ray PRO 250-30 and 270-30) and two tunable dye lasers (Sirah Lasertechnik, Cobra-Stretch). Detailed parameters are listed in Supplementary Table 17. The produced VUV light was spatially separated from other laser beams via a biconvex lithium fluoride lens (Korth Kristalle, R1 = R2 = 131 mm) in an off-axis geometry and passed 2 mm above the ice surface to ionize the subliming molecules. The VUV flux was monitored by a Faraday cup during TPD and was used to correct for variations of the TPD profiles throughout each experiment[7]. The resulting ions from VUV photoionization were mass-analyzed through reflectron time-of-flight mass spectrometry and detected by a dual microchannel plate (MCP) detector (Jordan TOF Products). The ion signals were then amplified with a preamplifier (Ortec, 9305), discriminated, and recorded by a multichannel scaler (FAST ComTec, MCS6A). For each recorded mass spectra, the ion arrival time and the accumulation time of ion signals were 3.2 ns accuracy and 2 min (3600 sweeps), respectively. An additional experiment was carried out without electron irradiation (blank) at 11.10 eV for CO–CH$_3$CH$_2$OH ice, and no sublimation event at $m/z = 74$ was observed. The gas phase mass spectra were collected for background gases in the main chamber and ethanol samples at 11.10 eV; no contamination molecules were detected (Supplementary Fig. 14). To assist in the identification of specific molecules, blank experiments were performed at 11.10 eV by adding 1% of ethyl formate (C$_2$H$_5$OCHO; Sigma Aldrich, >97%) or 1% of hydroxyacetone (HOCH$_2$C(O)CH$_3$, Thermo Scientific Chemicals, 95%) into the pre-mixed CO–CH$_3$CH$_2$OH gas mixture (Supplementary Table 16).

## Computational

Calculations were performed using the CBS–QB3 composite scheme to obtain accurate values for the energies of the neutral and cationic states of each species. This approach allows obtaining molecular parameters and energies with an accuracy of $0.01-0.02$ Å for bond lengths, $1-2°$ for bond angles, and $4-8$ kJ mol$^{-1}$ for relative energies. All ab initio calculations of the electronic structure were performed with the GAUSSIAN 09 package[76]. Five backbone isomers of reaction products were identified, differing in the position of the methyl, ethyl, carbonyl, and hydroxyl groups, namely lactaldehyde (7), 3-hydroxypropanal (18), ethyl formate (19), 1,3-propenediol (20), and 1,2-propenediol (21). For each backbone isomer, all possible isomers produced by rotation around selected C–O or C–C single and double bonds (subsequently referred to as conformers) were considered to obtain reliable ionization potential data. For this purpose, a computer program was developed to automatically prepare GAUSSIAN 09 input files for all possible conformers. The most energetically favorable product was found to be ethyl formate (19). The calculated conformer energies relative to ethyl formate (19) and adiabatic ionization potentials are shown in Fig. 3 and agree well with previous results (Supplementary Table 18). The same CBS-QB3 method was used for the optimization of transition states (TS), which are involved in the potential energy surfaces leading to 1,2-propenediol (21) and hydroxyacetone (22) from lactaldehyde (7). Initial geometries were chosen to resemble the proposed TS structures, and optimization was run to obtain the TS geometry. After optimization, TS was confirmed by inspection of normal mode eigenvalues and via internal reaction coordinate calculations to yield the correct stable products on either side of the barrier. The Cartesian coordinates, harmonic frequencies, and infrared intensities of the computed structures are available in Supplementary Data 1.

## Data availability

The data generated in this research are provided in the article, Supplementary Information, and Supplementary Data file, and are available upon request from the corresponding author(s). Source data are provided with this paper.

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

## Acknowledgements

The Hawaii group acknowledges support from the U.S. National Science Foundation (NSF), Division of Astronomical Sciences (AST-2403867). The University of Hawaii at Manoa and the W.M. Keck Foundation financed the construction of the experimental setup.

## Author contributions

R.I.K. directed the overall project. J.W., C.Z., and J.H.M. performed experiments. J.W. performed the data analyses. M.M.E., O.V.K., and I.O.A. carried out theoretical analysis. I.O.A. supervised the calculations. J.W., I.O.A., and R.I.K. wrote the manuscript, which was read, revised, and approved by all co-authors.

## Competing interests

The authors declare no competing interests.
