## [Transparent Peer Review file · Nature Communications]

Interstellar Formation of Lactaldehyde, A Key Intermediate in the Methylglyoxal Pathway

Corresponding Author: Professor Ralf Kaiser

Version 0:

Reviewer comments:

Reviewer #1

(Remarks to the Author)

The manuscript by Wang et al. describes some interesting experimental results on the formation of lactaldehyde on ices composed of CO and ethanol. Although the experimental work itself appears reasonable and accurate, the basic premise of the study of “interstellar formation” is questionable. The ice mixture chosen for the study of CO and ethanol (CH₃CH₂OH) is not realistic. Ethanol is a trace component of interstellar gas, with a typical abundance ratio relative to CO of CH₃CH₂OH/CO ~ 10⁻⁶ - 10⁻⁴; see for example, Requena-Torres et al. A&A 455, 971, 2006; Agundez et al. A&A 693, A34, 2023. The ice mixture created utilized a far higher amount of ethanol in a 50 torr/20 torr ratio of CO: CH₃CH₂OH, or 5:2. This sort of mixture is not representative of interstellar abundances. Further, it is highly unlikely that any interstellar ice would be composed of only CO and ethanol, considering the other interstellar compounds present such as HCN, NH₃, H₂CO, CH₃OH, etc., which are far more abundant than ethanol. Therefore, the experiment is really not “modeling interstellar conditions.”

There are other problems with the manuscript. The introduction is missing references. For example, the “formose” reaction was initially proposed by Breslow in a famous paper in 1959 (Tetrahedron Lett., 1(21), 22), not by Benner et al. in 2012. The second paragraph seems to be confused between hard-core biochemistry conducted under solution-phase, laboratory conditions and pre-biotic chemistry, which takes place in a completely different environment.

It is recommended that the work be published elsewhere and not in Nature Communications. The study seems more appropriate for the physical chemistry literature, with a large reduction of the interstellar emphasis.

Reviewer #2

(Remarks to the Author)

The manuscript makes a significant contribution to the field of astrochemistry and is scientifically rigorous. However, to further strengthen and quantify aspects of the text there are some questions regarding the methodology and discussion that should be clarified before this manuscript can be accepted for publication.

1. I believe the name of one of the co-authors has been misspelt on the title page of the manuscript (Mikhail M. Evseev vs. Milhail M. Evseev).
2. 5K is a low temperature that may accurately represent cold molecular cores in the interstellar medium, and is certainly lower than the 10-20 K temperatures that are more traditionally used in laboratory astrochemistry experiments. Could the authors provide more information as to why they chose this specific temperature? This is very important given that radiation chemistry reactions are often associated with strong temperature effects.
3. In the “Methods” section, the authors state that they made use of band strength constants to determine the actual composition of the deposited CO:EtOH ice, and have cited the band strengths measured by the bibliographic study of Bouilloud and co-workers. However, this study did not provide or measure any band strength constants for EtOH, and so the authors should specify from where the cited band strength constant was taken. Furthermore, the study of Bouilloud and co-workers measured the band strength constant for CO (which was then cited

by the present study) using transmission absorption IR spectroscopy (TAIRS). The authors have made use of reflection absorption IR spectroscopy (RAIRS) in their present study. It is incorrect to assume that a band strength measured using TAIRS is applicable to RAIRS studies. Indeed, there have been a number of studies (most prominently by the Leiden group; see, e.g., Ioppolo et al. *Rev. Sci. Instrum.* 84, 073112, 2013, and subsequent papers) that have emphasised that the use of band strength constants measured through TAIRS in RAIRS experiments is not to be recommended. This has ramifications for the accuracy of the calculated ice composition and should be commented upon/discussed.

4. Following on from 3, information on the density and refractive index of EtOH is given in the "Methods" section, but this information appears to be lacking for EtOD-D6. Indeed, the paper cited by the authors regarding the density of EtOH is that by Hudson, who did not consider EtOD-D6 in his study. The authors should quote the refractive index, density, and band strength of EtOD-D6 (along with appropriate references) and state whether the ice thickness and composition is different for the case of deuterated EtOH versus non-deuterated EtOH.
5. Could the authors provide any further information as to how stable their electron beam was and thus the fluences used, and if any precautions were taken to ensure the homogeneous irradiation of the target substrate / ice (e.g. rastering across the surface).
6. Was there any evidence of charging of the surface during irradiation?
7. The detection of the enol 1,3-propenediol (20) is very interesting. Is it possible to distinguish which stereoisomer (i.e., E- or Z-) is produced? I understand that there may be experimental limitations to this, but perhaps theory may help in this regard?
8. The product molecules formed through reactions 5-7 involve relatively large radicals. The authors should discuss whether this chemistry is likely to proceed via non-diffusive radical recombination reactions, as discussed by Jin and Garrod (*Astrophys. J. Suppl. Ser.* 249, 26, 2020). This discussion should consider the temperature at which this study was conducted, and should also speculate on whether alternative mechanisms (e.g., diffusive reactions) can begin to compete at higher temperatures.
9. The authors state that their 5 keV electron irradiations simulate the secondary electrons that are released by galactic cosmic rays as they interact with interstellar ices. However, there is significant literature to suggest that the most efficient secondary electrons at inducing chemistry in low temperature molecular solids are those of a lower energy, typically < 25 eV (see work by, e.g., Mason et al. *Faraday Discuss.* 168, 235-247, 2014; Boyer et al. *Surf. Sci.* 652, 26-32, 2016; Wu et al. *ACS Earth Space Chem.* 8, 79-88, 2023). Could the authors discuss a little more why they consider these 5 keV electrons are a good proxy for secondary electrons released along the track of a galactic cosmic ray?
10. Interstellar ices are usually (though admittedly not exclusively) dominated by H₂O. Could the authors discuss whether the presence of H₂O would be expected to affect the formation of the observed COMs, either positively or negatively?

Reviewer #3

(Remarks to the Author)

Reviewer #4

(Remarks to the Author)

I find the main thrust is chemistry, which is far away from my expertise, thus I do not want to give a formal review. However, here are a few thoughts on the manuscript.

The paper gets its main thrust from the idea that biological relevant molecules for the origin of life were formed in space, and delivered to Earth to start life here. This is a possibility, but the majority in the „origin of life“ research community considered the origin deep in the ocean at the black smokers as the most likely scenario. Thus, the authors should acknowledge that by citing, for example:

Camprubí, E. et al. The emergence of life. *Space Sci. Rev.* 215, 56 (2019).

Rimmer, P. B. & Shorttle, O. Origin of life's building blocks in carbon- and nitrogen-rich surface hydrothermal vents. *Life* 9, 12 (2019).

Deamer, D. Where did life begin? Testing ideas in prebiotic analogue conditions. *Life* 11, 134 (2021).

Substituting Galactic Cosmic Rays (GCR), i.e., mostly protons of GeV energy and more by 5 keV electrons is a far stretch, even though the authors have some arguments for that. These energetic protons do far more than the electrons, especially they produce radiation damage in addition to radiolytic formation. Look at literature of Mars surface or Europa ice for bio-relevant molecules surviving high energy radiation.

Reviewer #5

(Remarks to the Author)

The manuscript NCOMMS-24-29938 presents the first conclusive laboratory and computational evidence for the interstellar formation of lactaldehyde and other complex organic molecules that are key to the origins of life on Earth. Being of high impact in the fields of astrochemistry, planetary science, and astrobiology, the manuscript is well written and contains all the information including excellent figures needed to support conclusions. The laboratory techniques applied here are state-of-the-art and enable the authors to detect and trace newly formed species with unprecedented sensitivity. Therefore, I highly recommend the manuscript for publication in Nature Communications. My one and only comment to the authors is that I found the introduction a bit hard to follow. Some of the text seems to better belong into the results and discussion sections. I would therefore suggest the authors to amend the first section of the paper such that the topic is well introduced and its importance highlighted without necessarily going into much details concerning specific formation and destruction reaction pathways. For instance, here a more astronomy-oriented introduction could be beneficial to the manuscript. I hope this helps further improving an already excellent piece of work.

Reviewer #6

(Remarks to the Author)

The manuscript "Interstellar Formation of Lactaldehyde (CH₃CH(OH)CHO)-A Key Intermediate in the Methylglyoxal Pathway" by Wang et al. describes the formation of intermediate bioessential organic compound "lactaldehyde" in carbon monoxide and ethanol ice mixture under simulated interstellar conditions. Using isomer-selective photoionisation reflection TOF MS, the study claimed the detection of acetaldehyde and its three isomer compounds 3-hydroxypropanal (HOCH₂CH₂CHO), ethyl formate (CH₃CH₂OCHO), and 1,3-propenediol (HOCH₂CHCHOH) in the gas phase. The study also revealed formation pathways of aldehydes in the interstellar environments.

On page six, starting from the line 119, MS related findings are given. The study focused on four photons energies 11.10 eV, 10.23 eV, 9.71 eV, and 9.29 eV to distinguish lactaldehyde isomers that were formed through radical-radical recombination.

Comments:

Line 127-129:

Authors used a method to replace "H" with "D" in the molecule to confirm that the signal at m/z 74 means that the molecule should contain 6 hydrogens and hence verify the molecular formula C₃H₆O₂. In this way, they ruled out other isomers compounds. I strongly suggest that authors should extend this in the method or discussion to provide a further better understanding for the reader.

Secondly, how about other organic compounds with molecular weight 74 and have 6 hydrogens in the molecule. For example, C₃H₆S, C₂H₆N₂O, CH₆N₄. I suggest authors should add a comparative case study for such compounds as well to further strengthen their findings.

Methods/Experimental:

As this work offers an identification of species at m/z 74 (aldehyde containing six hydrogen), this is utmost important to talk about the contamination/decontamination of the experiment. I could not find that authors discussed this aspect. For example, there could be contaminant from previous usage of the system/apparatus or contaminant within sample itself that could result in such species that produce a signal at m/z 74. The same is the situation with the "D" containing species that result signal at m/z 80 in this case. For example certain species with mol. weight 74 could also give the signal that authors observed here: C₄H₁₀O, C₃H₁₀N₂, C₃H₁₀Si, C₃H₃Cl, C₂H₂O₃. Similarly, for m/z 80, one could have C₆H₈, C₄H₄N₂, C₃H₆F₂, C₂H₅ClO, C₂H₅ClO, C₂H₂ClF, CH₄O₂S, HBr. Therefore, I suggest to address this issue as well in a substantial way. An addition of a table would be good in the Supplementary Material for better understand and the confidence on the findings.

Version 1:

Reviewer comments:

Reviewer #1

(Remarks to the Author)

My main objection to this manuscript has been the implied idea that somehow the experiments, which are of high quality in terms of physical chemistry, are simulating interstellar conditions. They do not. The authors have subsequently explained that indeed they do not represent true interstellar conditions, and the experiments have been performed to gain insight into fundamental reaction processes that may be applicable to interstellar chemistry and other venues. They have added some qualifying comments to the text in this regard. I therefore recommend publication, with one caveat. The term "Origins of Life" appears often in the text and in capital letters and italics. Although this work is distantly related to the origin of life, its repeated emphasis seems inappropriate and should be less quoted in the text.

Reviewer #2

(Remarks to the Author)

The authors have addressed all the points we raised and have made appropriate additions to the manuscript

Reviewer #3

(Remarks to the Author)

Reviewer #4

(Remarks to the Author)

The authors addressed my concerns adequately.

Reviewer #6

(Remarks to the Author)

The authors have reasonably addressed my comments and adapted suggestions.

Response to Reviewers

Reviewer #1 (Remarks to the Author):

The manuscript by Wang et al. describes some interesting experimental results on the formation of lactaldehyde on ices composed of CO and ethanol.

Reply: We are glad to hear that the referee found our results interesting.

Although the experimental work itself appears reasonable and accurate, the basic premise of the study of “interstellar formation” is questionable.

Reply: Laboratory experiments on the interaction of galactic cosmic rays and their secondary electrons with interstellar analog ices can never simulate the complexity of interstellar environments with such as a wide range of kinetic energies of the irradiating particles and the ultraviolet (UV) irradiation field as well as the composition of the ice targets themselves. Here, our experiments have to be designed to first understand the physical and chemical processes for the formation pathways of C₃H₆O₂ isomers, using relatively simple model systems such as binary carbon monoxide and ethanol (CO–CH₃CH₂OH) ices under controlled conditions. These simulation experiments lay the groundwork before expanding the experiments to more complex systems. The derived reaction mechanisms from the CO–CH₃CH₂OH ices, however, present versatile pathways to form more complex organic molecules in these low temperature environments and knowledge can be applied without restrictions to extraterrestrial ices. Since the investigation of the CO–CH₃CH₂OH model ices has been completed, the next step would be to investigate the interaction of cosmic ray particles and their secondary electrons with *realistic ice mixtures* by incorporating water (H₂O), carbon dioxide (CO₂), and methanol (CH₃OH), and so on.

The ice mixture chosen for the study of CO and ethanol (CH₃CH₂OH) is not realistic. Ethanol is a trace component of interstellar gas, with a typical abundance ratio relative to CO of CH₃CH₂OH/CO ~ 10⁻⁶ -10⁻⁴; see for example, Requena-Torres et al. A&A 455, 971, 2006; Agundez et al. A&A 693, A34, 2023. The ice mixture created utilized a far higher amount of ethanol in a 50 torr/20 torr ratio of CO: CH₃CH₂OH, or 5:2. This sort of mixture is not representative of interstellar abundances.

Reply: We would like to point out that this study focuses on the reaction mechanisms *in low-temperature ices* rather than the reactions in the gas phase. The abundance ratio of CO to ethanol in the ices can be very different than that of the gas phase. *First*, CO in ice starts to desorb at around 20 K, which may lead to a low abundance of CO in the ices under some interstellar environments with a higher temperature. *Second*, ethanol is believed to be efficiently formed on dust grains, which is also mentioned in the reference that the reviewer mentioned (M. A. Requena-Torres¹ et al. A&A 455, 971, 2006). Once formed, due to the low temperature, ethanol molecules can stay on those dust grains. In fact, based on recent observations from James Webb Space Telescope (JWST), W. R. M. Rocha et al. reported a statistically robust detection of ethanol in the ices with abundances of above 1% with respect to water towards one low- and one high-mass protostar (W. R. M. Rocha et al. A&A, 683, A124 (2024)).

Third, as mentioned above, this work is a proof of concept study focusing on mechanistic information regarding the formation of C₃H₆O₂ isomers including lactaldehyde and its isomers. These ices are not intended to mimic the exact levels of ethanol within interstellar ice grains, but rather are used as a model to extract pertinent information regarding possible chemical reactions. Normally, when conducting experiments with different ratios of reactants in the ice, we find an overall decrease of reaction products, but no influence on branching ratios of isomers as long as all products formally require the same amount of reactant molecules. The non-equilibrium chemistry studied here is a chemistry of opportunity – whenever two radicals or excited state molecules meet in a favorable geometry, they will react, even if thermodynamically more favorable channels are available.

Furthermore, although the ratio of CO to CH₃CH₂OH chosen in our experiments may not be a typical abundance ratio observed in molecular clouds, this ratio ensures the highest possible yield of products and thus facilitates their detection. Given the rather low signal associated with lactaldehyde, the higher ratios of CO to CH₃CH₂OH would render us unable to distinguish the ion signal from noise. We have added the following sentences in the Experimental section as highlighted in yellow (the end of Page 12):

“Although the ratio of carbon monoxide to ethanol used in the experiments may not be a typical abundance ratio observed in molecular clouds, this ratio ensures the highest possible yield of C₃H₆O₂ isomers and thus facilitates their detection.”

Further, it is highly unlikely that any interstellar ice would be composed of only CO and ethanol, considering the other interstellar compounds present such as HCN, NH₃, H₂CO, CH₃OH, etc., which are far more abundant than ethanol. Therefore, the experiment is really not “modeling interstellar conditions.”

Reply: It is important to note that no simulation experiment can replicate the diverse environment of the interstellar medium and the chemical complexity concurrently. Therefore, simulation experiments have to be conducted first with well-defined model ices. Our previous publications, as well as the literature, show that complex organic molecules are often formed from key precursors, which, in turn, can be synthesized from several distinct ice mixtures. Here, the CO–CH₃CH₂OH ices selected in our laboratory simulations present a model ice to understand the formation pathways of lactaldehyde and its isomers *in a comprehensive way*. The complexity of the detected irradiation products justifies this simple model. *Only after understanding the processes and products in processed CO–CH₃CH₂OH ices as presented here can we extend our studies to the formation mechanisms of lactaldehyde and its isomers in more complex and realistic systems*, such as those incorporating water (H₂O), carbon dioxide (CO₂), and methanol (CH₃OH). Ice mixtures including HCN may also be considered; however, to the best of our knowledge, HCN has not been confirmed to be present in interstellar ices so far. Recent JWST searches for HCN ices also failed (McClure et al. Nat. Astron 7:431-443 (2023)). We have added the following sentences in the Discussion section (Page 9):

“The CO–CH₃CH₂OH ices selected in our laboratory simulations present a model ice to understand the formation pathways of lactaldehyde and its isomers in a comprehensive way. Future experiments can explore the effects of ice composition by incorporating other simple molecules common to interstellar ices such as water (H₂O), carbon dioxide (CO₂), and methanol (CH₃OH) into the ice mixture.”

In addition, we removed the “modeling interstellar conditions” in the abstract and revised the sentence as follows (Page 2):

“Here, we unveil the formation of lactaldehyde (CH₃CH(OH)CHO) by barrierless recombination of formyl (HĊO) and 1-hydroxyethyl (CH₃ĊHOH) radicals in interstellar ice analogs composed of carbon monoxide (CO) and ethanol (CH₃CH₂OH).”

There are other problems with the manuscript. The introduction is missing references. For example, the “formose” reaction was initially proposed by Breslow in a famous paper in 1959 (Tetrahedron Lett., 1(21), 22), not by Benner et al. in 2012.

Reply: Thank you for pointing it out. We have added this reference in the introduction.

The second paragraph seems to be confused between hard-core biochemistry conducted under solution-phase, laboratory conditions and pre-biotic chemistry, which takes place in a completely different environment.

Reply: We now only keep the text for the prebiotic chemistry in the second paragraph and have moved the rest into the Discussion section. Meanwhile, the compound numbers have been updated accordingly.

It is recommended that the work be published elsewhere and not in Nature Communications. The study seems more appropriate for the physical chemistry literature, with a large reduction of the interstellar emphasis.

Reply: We are grateful toward the reviewer for their hard work and suggestions that have undoubtedly improved the quality of this manuscript. However, we respectfully disagree with the reviewer’s assessment. *Our results of the interstellar formation of the lactaldehyde and its isomer as well as their synthesis mechanisms reveal fundamental formation pathways for complex, biologically relevant aldehydes in interstellar environments. These findings are exceptionally significant to a large number of disciplines (astrochemistry, organic chemistry, physical chemistry, and astrobiology) and will certainly grab the wide-reaching attention of a broad audience.*

Reviewer #2 (Remarks to the Author):

The manuscript makes a significant contribution to the field of astrochemistry and is scientifically rigorous.

Reply: Thank you very much for the careful review of our work and for providing us with suggestions. We are proud to receive the reviewer's commendation for the quality and novelty of our work.

However, to further strengthen and quantify aspects of the text there are some questions regarding the methodology and discussion that should be clarified before this manuscript can be accepted for publication.

Reply: We have clarified the methodology and discussion below according to your concerns. All changes are highlighted in yellow.

1. I believe the name of one of the co-authors has been misspelt on the title page of the manuscript (Mikhail M. Evseev vs. Milhail M. Evseev).

Reply: Thank you for pointing it out. We have corrected it.

2. 5K is a low temperature that may accurately represent cold molecular cores in the interstellar medium, and is certainly lower than the 10-20 K temperatures that are more traditionally used in laboratory astrochemistry experiments. Could the authors provide more information as to why they chose this specific temperature? This is very important given that radiation chemistry reactions are often associated with strong temperature effects.

Reply: At a low temperature of 5 K, radicals and species are expected to exhibit more limited mobility. Such cold ices are able to preserve these reactive intermediates intact to provide valuable mechanistic information. This is important to the goal of current work, which is aiming at the formation mechanisms of C₃H₆O₂ isomers in irradiated ices. Although the temperatures of 5 K used in these experiments are slightly lower than that typically found in molecular clouds, the low temperatures employed in these experiments are similar to the theorized minimum temperature of cold starless cores, 6 K (Juvela et al., *Astrophys. J.* 2011, 739, 63), as they approach thermal equilibrium with the cosmic microwave background (2.7 K). No experiment can fully replicate the exact conditions in interstellar clouds, and even within clouds the temperatures can vary,

potentially also to lower temperatures than 10 K. In addition, as pure CO ice starts to desorb at around 20 K, a lower temperature for performing the experiment is preferred. To clarify the utility of performing these experiments at 5 K, we have added the following sentences on Page 12:

“Although the temperatures of 5 K used in these experiments are slightly lower than that typically found in molecular clouds, intact reactive intermediates can be preserved to provide valuable mechanistic insights in such cold ices.”

and on Page 5:

“Due to the limited molecular mobility at 5 K, these radicals are preserved within the ice; radicals may not be able to react if they form without nearby radicals⁴³.”

3. In the “Methods” section, the authors state that they made use of band strength constants to determine the actual composition of the deposited CO:EtOH ice, and have cited the band strengths measured by the bibliographic study of Bouilloud and co-workers. However, this study did not provide or measure any band strength constants for EtOH, and so the authors should specify from where the cited band strength constant was taken. Furthermore, the study of Bouilloud and co-workers measured the band strength constant for CO (which was then cited by the present study) using transmission absorption IR spectroscopy (TAIRS). The authors have made use of reflection absorption IR spectroscopy (RAIRS) in their present study. It is incorrect to assume that a band strength measured using TAIRS is applicable to RAIRS studies. Indeed, there have been a number of studies (most prominently by the Leiden group; see, e.g., Ioppolo et al. Rev. Sci. Instrum. 84, 073112, 2013, and subsequent papers) that have emphasised that the use of band strength constants measured through TAIRS in RAIRS experiments is not to be recommended. This has ramifications for the accuracy of the calculated ice composition and should be commented upon/discussed.

Reply: We performed independent IR experiments to collect the FTIR spectra of *pure* ethanol ices at 5 K *with known thickness*. In detail, IR spectra were collected for ethanol (760 ± 50 nm), ethanol- d_6 (810 ± 50 nm), and ethanol- $^{13}C_2$ (450 ± 50 nm) ices. Comparing the integrated area of multiple absorption bands of these pure ethanol ices with that of mixture ices, the thickness of the deposited ethanol, ethanol- d_6 , or ethanol- $^{13}C_2$ in the mixture ices can be determined. Then the column densities of ethanol molecules in mixture ices can be estimated based on their densities and masses.

Therefore, the band strength constants for ethanol molecules are not necessarily needed for our purpose. To clarify this, we have added the following sentences in the Experimental section (Page 12):

“FTIR spectra of pure ethanol, ethanol-d₆, and ethanol-¹³C₂ ices were collected at 5 K with thicknesses of 760 ± 50 nm, 810 ± 50 nm, and 450 ± 50 nm, respectively (Supplementary Figs. 11–13), which were used to determine the thicknesses and column densities of ethanol, ethanol-d₆, and ethanol-¹³C₂ in the mixture ices based on the integrated area of multiple absorption bands.”

We have commented on the calculated ice composition by adding the following sentences (Page 12):

“It is necessary to note that the absorption coefficients of carbon monoxide were obtained via transmission absorption IR spectroscopy, which may differ from those obtained using reflection absorption IR spectroscopy^{73,74}. Other factors, such as the thickness of the ices and the angle of incidence of the IR beam, can affect the relative peak heights in reflectance IR spectra. Here, we use the absorption coefficients in reflection as a means to estimate the ratio of components in the ice mixtures.”

In addition, we have revised the sentence as follows:

“..., the ratio of carbon monoxide to ethanol in the ice mixture (CO–CH₃CH₂OH) was *estimated* to be $2.5 \pm 0.4:1$ (Supplementary Table 16).”

4. Following on from 3, information on the density and refractive index of EtOH is given in the “Methods” section, but this information appears to be lacking for EtOD-D6. Indeed, the paper cited by the authors regarding the density of EtOH is that by Hudson, who did not consider EtOD-D6 in his study. The authors should quote the refractive index, density, and band strength of EtOD-D6 (along with appropriate references) and state whether the ice thickness and composition is different for the case of deuterated EtOH versus non-deuterated EtOH.

Reply: As mentioned above, the band strength constants for ethanol-d₆ are not necessarily needed. As in general the difference between heavier isotopologues and standard isotopologues in the indexes is marginal, we assume the index of the standard isotopologue is the same for heavier isotopologues. For instance, H₂O and D₂O differ in index of refraction by 0.5% (H. Odhner et al.

J. Chem. Eng. Data, 57, 166–168 (2012)). For isotopically labeled ice mixtures, variations in density were considered based on the masses of reactants. The experimental condition table (Supplementary Table 16) including ice composition and thickness was updated. We have revised the sentences to clarify it in the Experimental section (Page 12):

“The average index of 1.26 ± 0.04 was used to derive the thickness of the mixture ice (CO–CH₃CH₂OH) from the refractive indexes of carbon monoxide ice ($n = 1.25 \pm 0.03$)³⁴ and that of ethanol ice ($n = 1.26$)⁷². This average index was used for the isotopically labeled ice mixtures. The ice thicknesses of CO–CH₃CH₂OH ice were determined to be 880 ± 50 nm. The densities of 0.80 ± 0.01 g cm⁻³ for CO ice³⁴ and 0.584 g cm⁻³ for CH₃CH₂OH ice⁷² were used. For isotopically labeled ice mixtures, variations in density were considered based on the masses of reactants.”

5. Could the authors provide any further information as to how stable their electron beam was and thus the fluences used, and if any precautions were taken to ensure the homogeneous irradiation of the target substrate / ice (e.g. rastering across the surface).

Reply: We have taken specific precautions to ensure the stability of the electron beam and homogeneous irradiation of the ices in the experiments:

Before each irradiation experiment, the electron gun was operated for at least one hour to stabilize its current, which was monitored using a Faraday cup that can be inserted into the electron beam path. The incident electron current was measured both before and after irradiation, with fluctuations kept less than 3 nA within 5 or 10 minutes of irradiation. Additionally, prior to irradiation, a phosphor screen, of which the center is mounted around 30 mm above the center of the sample substrate, was moved to the irradiation position to monitor and adjust the electron beam, ensuring uniform exposure across the sample substrate as the beam is rastered over the surface. We have added the following sentences in the Experimental section (Page 13):

“Prior to irradiation, a phosphor screen was used to monitor and adjust the electron beam, ensuring uniform exposure across the sample substrate. The electron beam current was measured using a Faraday cup before and after irradiation, with fluctuations kept within 3 nA.”

6. Was there any evidence of charging of the surface during irradiation?

Reply: No, we have no evidence of charging of the surface during irradiation. The ices were deposited onto a polished silver substrate, which is grounded.

7. The detection of the enol 1,3-propenediol (20) is very interesting. Is it possible to distinguish which stereoisomer (i.e., E- or Z-) is produced? I understand that there may be experimental limitations to this, but perhaps theory may help in this regard?

Reply: Indeed, the chemistry of enols in the ISM is very interesting and has attracted increasing attention. To distinguish which stereoisomer of the enol 1,3-propenediol is produced in our experiments, isomer-specific spectroscopic techniques such as photolysis and/or photoionization efficiency (PIE) measurements may help. However, 1,3-propenediol has 13 stereoisomers (8 *syn*-conformers and 5 *anti*-conformers), and their computed ultraviolet–visible spectra (for photolysis experiment) and PIE curves (for PIE experiment) can complicate the interpretation of the experimental data. Therefore, it could be very difficult to distinguish them and is beyond the scope of this study.

8. The product molecules formed through reactions 5-7 involve relatively large radicals. The authors should discuss whether this chemistry is likely to proceed via non-diffusive radical recombination reactions, as discussed by Jin and Garrod (*Astrophys. J. Suppl. Ser.* 249, 26, 2020). This discussion should consider the temperature at which this study was conducted, and should also speculate on whether alternative mechanisms (e.g., diffusive reactions) can begin to compete at higher temperatures.

Reply: Under the low-temperature of 5 K in our study, radicals remain relatively immobile, preventing significant diffusion-driven recombination. Non-diffusive reactions can occur when radicals are formed in close proximity to each other, allowing the formation of product molecules via radical-radical recombination mechanisms. This is especially relevant for relatively large radicals such as $\text{CH}_3\dot{\text{C}}\text{HOH}$, $\dot{\text{C}}\text{H}_2\text{CH}_2\text{OH}$ and $\text{CH}_3\text{CH}_2\dot{\text{O}}$, indicating that the formation of lactaldehyde and its isomers is likely to proceed via non-diffusive radical recombination mechanisms. This is suggested by the interesting work that the reviewer mentioned. At higher temperatures such as 20 K, however, diffusive mechanisms may compete with the non-diffusive pathways as radicals such as $\text{H}\dot{\text{C}}\text{O}$ become more mobile. (Jin and Garrod, *Astrophys. J. Suppl. Ser.* 249, 26, 2020 and Garrod et al. *The Astrophys. J.* 682, 283–302, 2008). We have added the following discussion on Page 8:

“Due to the limited molecular mobility caused by the low temperatures of 5 K, these radicals are preserved within the ice; radicals may not be able to react if they form without nearby radicals⁴³. This is especially relevant for relatively large radicals such as **17**, **22** and **23**, indicating that the formation of lactaldehyde and its isomers is likely to proceed via non-diffusive radical recombination mechanisms⁵². However, at higher temperatures such as 20 K, diffusive mechanisms may compete with the non-diffusive pathways as radicals such as HĈO become more mobile^{52,53}.”

9. The authors state that their 5 keV electron irradiations simulate the secondary electrons that are released by galactic cosmic rays as they interact with interstellar ices. However, there is significant literature to suggest that the most efficient secondary electrons at inducing chemistry in low temperature molecular solids are those of a lower energy, typically < 25 eV (see work by, e.g., Mason et al. Faraday Discuss. 168, 235-247, 2014; Boyer et al. Surf. Sci. 652, 26-32, 2016; Wu et al. ACS Earth Space Chem. 8, 79-88, 2023). Could the authors discuss a little more why they consider these 5 keV electrons are a good proxy for secondary electrons released along the track of a galactic cosmic ray?

Reply: The electron kinetic energy of 5 keV was used in the present experiments because its linear energy transfer is comparable to that of 10–20 MeV GCR protons deposit into ices. In addition, it is likely that 5 keV electrons also produce low energy (<25 eV) secondary electrons just like GCRs do. We have added discussion sentences by mentioning the above studies in the Supplementary Information as follows (Page S4):

“GCRs primarily lose energy through ionization of the target molecules in the ice and generate secondary electrons that can induce further ionization, resulting in electron cascades⁵. Consequently, the kinetic energy of the resulting electrons are in ranges of a few eV up to 10 keV depending on the energy of the GCR particle^{6,7}. These electrons, especially for low-energy (< 20 eV) secondary electrons, could be a significant contributor to the interstellar synthesis of prebiotic molecules⁷⁻⁹. Therefore, the chemical effects of GCRs on ices can be simulated by irradiating the ices with energetic electrons as a proxy^{10,11}. The present experiments utilized the electron kinetic energies of 5 keV as their linear energy transfer is similar to that of 10–20 MeV GCR protons deposit into ices^{12,13}. In addition, 5 keV electrons have been widely used previously to simulate the secondary electrons released during GCRs penetrating interstellar ices^{10,14-16}.”

10. Interstellar ices are usually (though admittedly not exclusively) dominated by H₂O. Could the authors discuss whether the presence of H₂O would be expected to affect the formation of the observed COMs, either positively or negatively?

Reply: Normally, when conducting experiments diluting the ice with water, we found an overall decrease of reaction products, but no influence on branching ratios of isomers as long as all products formally require the same amount of reactant molecules. The non-equilibrium chemistry studied here is a chemistry of opportunity – whenever two radicals or excited state molecules meet in a favorable geometry, they will react, even if thermodynamically more favorable channels are available. In this sense, adding water to our sample would mainly change quantitative results. Given the rather low signal associated with lactaldehyde, more realistic ratios or dilution with water would render us unable to distinguish the ion signal from noise. Since the investigation of the CO–CH₃CH₂OH model ices has been completed, the next step would be to investigate *realistic ice mixtures* by incorporating interstellar ice molecules such as water (H₂O). We have added the following sentences in the Discussion section (Page 9):

“Future experiments can explore the effects of ice composition on these molecules by incorporating other simple molecules common to interstellar ices such as water (H₂O), carbon dioxide (CO₂), and methanol (CH₃OH) into the ice mixture.”

Reviewer #3 (Remarks to the Author):

Reply: Thank you very much for your efforts in reviewing our work.

Reviewer #4 (Remarks to the Author):

I find the main thrust is chemistry, which is far away from my expertise, thus I do not want to give a formal review. However, here are a few thoughts on the manuscript.

Reply: Thank you for reviewing our work and providing us with suggestions to improve our manuscript.

The paper gets its main thrust from the idea that biological relevant molecules for the origin of life were formed in space, and delivered to Earth to start life here. This is a possibility, but the majority in the „origin of life“ research community considered the origin deep in the ocean at the black smokers as the most likely scenario. Thus, the authors should acknowledge that by citing, for example:

Camprubí, E. et al. The emergence of life. *Space Sci. Rev.* 215, 56 (2019).

Rimmer, P. B. & Shorttle, O. Origin of life's building blocks in carbon- and nitrogen-rich surface hydrothermal vents. *Life* 9, 12 (2019).

Deamer, D. Where did life begin? Testing ideas in prebiotic analogue conditions. *Life* 11, 134 (2021).

Reply: Thank you for the suggestion. We have cited the above references and added the sentence in the introduction (Page 3):

“Although the deep ocean hydrothermal vents are considered as a likely scenario for the emergence of life¹⁶⁻¹⁸, a substantial fraction of the prebiotic organic molecules on proto-Earth may have been of extraterrestrial origin¹⁹.”

Substituting Galactic Cosmic Rays (GCR), i.e., mostly protons of GeV energy and more by 5 keV electrons is a far stretch, even though the authors have some arguments for that. These energetic protons do far more than the electrons, especially they produce radiation damage in addition to radiolytic formation. Look at literature of Mars surface or Europa ice for bio-relevant molecules surviving high energy radiation.

Reply: Indeed, galactic and solar cosmic rays can account for the destruction of organic compounds on Mars surface or Europa ice. For instance, Pavlov et al. 2012 revealed that the preservation of ancient complex organic molecules in the shallow (~10 cm depth) subsurface of

rocks could be highly problematic (Pavlov, A. A., et al. *Geophys. Res. Lett.* 39, 2012). However, we would like to point out that the current simulation experiments were performed to unravel the formation pathways of complex organic molecules *in interstellar ices in cold molecular clouds* through the interaction with GCRs.

In addition, the energetic electrons are considered as a proxy of GCRs in laboratory simulation experiments and have been widely accepted in the astrochemistry community. The electron kinetic energy of 5 keV was used in the present experiments because its linear energy transfer (5.0 ± 0.5 keV μm^{-1}) is comparable to that of 10–20 MeV GCR protons (a few keV μm^{-1}) deposit into ices. We have added the following sentences to clarify it in the Supplementary Information (Page S4):

“The current simulation experiments were performed to unravel the formation pathways of complex organic molecules in interstellar ices in cold molecular clouds through the interaction with GCRs. The main constituents of GCRs are energetic protons (H^+) and helium nuclei (He^{2+})³. It is important to note that no laboratory experiment can directly mimic the interaction of energetic GCRs with ices due to the lack of experimental device that can generate a broad range (from MeV to the PeV) of kinetic energies of protons and helium nuclei⁴. However, the physical effects of GCRs interacting with ices are known. GCRs primarily lose energy through ionization of the target molecules in the ice and generate secondary electrons that can induce further ionization, resulting in electron cascades⁵. Consequently, the kinetic energy of the resulting electrons are in ranges of a few eV up to 10 keV depending on the energy of the GCR particle^{6,7}. These electrons, especially for low-energy (< 20 eV) secondary electrons, could be a significant contributor to the interstellar synthesis of prebiotic molecules⁷⁻⁹. Therefore, the chemical effects of GCRs on ices can be simulated by irradiating the ices with energetic electrons as a proxy^{10,11}. The present experiments utilized the electron kinetic energies of 5 keV as their linear energy transfer is similar to that of 10–20 MeV GCR protons deposit into ices^{12,13}. In addition, 5 keV electrons have been widely used previously to simulate the secondary electrons released during GCRs penetrating interstellar ices^{10,14-16}.”

Reviewer #5 (Remarks to the Author):

The manuscript NCOMMS-24-29938 presents the first conclusive laboratory and computational evidence for the interstellar formation of lactaldehyde and other complex organic molecules that are key to the origins of life on Earth. Being of high impact in the fields of astrochemistry, planetary science, and astrobiology, the manuscript is well written and contains all the information including excellent figures needed to support conclusions. The laboratory techniques applied here are state-of-the-art and enable the authors to detect and trace newly formed species with unprecedented sensitivity. Therefore, I highly recommend the manuscript for publication in Nature Communications.

Reply: Thank you very much for the careful review of our work. We are proud to receive the reviewer's commendation for the quality and novelty of our work.

My one and only comment to the authors is that I found the introduction a bit hard to follow. Some of the text seems to better belong into the results and discussion sections. I would therefore suggest the authors to amend the first section of the paper such that the topic is well introduced and its importance highlighted without necessarily going into much details concerning specific formation and destruction reaction pathways. For instance, here a more astronomy-oriented introduction could be beneficial to the manuscript. I hope this helps further improving an already excellent piece of work.

Reply: We have moved the texts for biochemistry in the second paragraph of the Introduction section to the Discussion section. Meanwhile, the compound numbers have been updated accordingly.

Reviewer #6 (Remarks to the Author):

The manuscript “Interstellar Formation of Lactaldehyde ($\text{CH}_3\text{CH}(\text{OH})\text{CHO}$)-A Key Intermediate in the Methylglyoxal Pathway” by Wang et al. describes the formation of intermediate bioessential organic compound “lactaldehyde” in carbon monoxide and ethanol ice mixture under simulated interstellar conditions. Using isomer-selective photoionisation reflection TOF MS, the study claimed the detection of acetaldehyde and its three isomer compounds 3-hydroxypropanal ($\text{HOCH}_2\text{CH}_2\text{CHO}$), ethyl formate ($\text{CH}_3\text{CH}_2\text{OCHO}$), and 1,3-propenediol ($\text{HOCH}_2\text{CHCHOH}$) in the gas phase. The study also revealed formation pathways of aldehydes in the interstellar environments.

On page six, starting from the line 119, MS related findings are given. The study focused on four photons energies 11.10 eV, 10.23 eV, 9.71 eV, and 9.29 eV to distinguish lactaldehyde isomers that were formed through radical-radical recombination.

Comments:

Reply: Thank you very much for the careful review of our work and for providing us with suggestions.

Line 127-129:

Authors used a method to replace “H” with “D” in the molecule to confirm that the signal at m/z 74 means that the molecule should contain 6 hydrogens and hence verify the molecular formula $\text{C}_3\text{H}_6\text{O}_2$. In this way, they ruled out other isomers compounds. I strongly suggest that authors should extend this in the method or discussion to provide a further better understanding for the reader.

Reply: To further confirm the formula, we have conducted additional low dose experiments using *fully* carbon-13 isotopically labeled ice ($^{13}\text{CO}-^{13}\text{CH}_3^{13}\text{CH}_2\text{OH}$) and the partially ^{18}O isotopically labeled ice ($\text{C}^{18}\text{O}-\text{CH}_3\text{CH}_2\text{OH}$) at 11.10 eV. The TPD profile of $m/z = 77$ in irradiated $^{13}\text{CO}-^{13}\text{CH}_3^{13}\text{CH}_2\text{OH}$ ice agrees nicely with that of $m/z = 74$ in irradiated $\text{CO}-\text{CH}_3\text{CH}_2\text{OH}$ ice (Fig. 5a), suggesting that the formula should contain *exactly* three carbon atoms. Recall that the fully deuterated ice ($\text{CO}-\text{CD}_3\text{CD}_2\text{OD}$) experiment confirms that the formula should contain *exactly* six hydrogen atoms. Therefore, given the molecular weights of the reactants in $\text{CO}-\text{CH}_3\text{CH}_2\text{OH}$ ice, the ion signal at $m/z = 74$ can only belong to $\text{C}_3\text{H}_6\text{O}_2$ isomers. This assignment was further confirmed by the additional experiment using $\text{C}^{18}\text{O}-\text{CH}_3\text{CH}_2\text{OH}$ ice, in which the three

sublimation events of $m/z = 76$ in irradiated $C^{18}O-CH_3CH_2OH$ ice match with that of $m/z = 74$ in irradiated $CO-CH_3CH_2OH$ ice (Supplementary Fig. 8), suggesting that the formula should contain at least one oxygen atom. We have revised the sentences as follows (Page 6):

“Given the molecular weights of the reactants in $CO-CH_3CH_2OH$ ice, the ion signal at *mass-to-charge* (m/z) of 74 can belong to organic compounds with formulae including C_6H_2 , $C_4H_{10}O$, $C_3H_6O_2$, and/or $C_2H_2O_3$. Hence it is imperative to confirm the molecular formula using isotopically labeled precursors. Utilizing the fully deuterated ices with $CO-CD_3CD_2OD$ ice, the TPD profile of $m/z = 80$ in irradiated $CO-CD_3CD_2OD$ ice matches well with that of $m/z = 74$ in irradiated $CO-CH_3CH_2OH$ ice (Fig. 5a), indicating the presence of exactly six hydrogen atoms. In addition, the TPD profile of $m/z = 74$ in irradiated $CO-CH_3CH_2OH$ ice was found to undergo an isotopic mass shift to $m/z = 77$ in fully carbon-13 isotopically labeled ice ($^{13}CO-^{13}CH_3^{13}CH_2OH$), confirming the inclusion of exactly three carbon atoms. Furthermore, the TPD profile of $m/z = 74$ in $CO-CH_3CH_2OH$ ice shifts two atomic mass units (amu) to $m/z = 76$ in $C^{18}O-CH_3CH_2OH$ ice (Supplementary Fig. 8), indicating the presence of at least one oxygen atom. These findings validate the assignment of the reaction products of the molecular formula $C_3H_6O_2$ for the TPD profile (Supplementary Note 1).”

Secondly, how about other organic compounds with molecular weight 74 and have 6 hydrogens in the molecule. For example, C_3H_6S , $C_2H_6N_2O$, CH_6N_4 . I suggest authors should add a comparative case study for such compounds as well to further strengthen their findings.

Reply: As mentioned above, the results of fully deuterated and fully carbon-13 isotopically labeled ices confirm the inclusion of *exactly* six hydrogen atoms and *exactly* three carbon atoms for the formula; the additional $C^{18}O-CH_3CH_2OH$ ice experiment suggests the formula should contain at least one oxygen atom. Hence, the molecular formula $C_3H_6O_2$ is confirmed for the $CO-CH_3CH_2OH$ system. In addition, our ice mixtures do not contain sulfur nor nitrogen, and thus C_3H_6S , $C_2H_6N_2O$, and CH_6N_4 are not possible. Furthermore, by lowering the photon energy to 9.29 eV, at which C_3H_6S isomers including methylthiirane (IE = 8.6 ± 0.2 eV), thietane (IE = 8.65 ± 0.01 eV), allyl mercaptan (IE = 9.25 eV), methyl vinyl sulfide (IE_{vertical} = 8.44 eV), and thioacetone (IE_{vertical} = 8.60 ± 0.05 eV) can be ionized; however, no sublimation events were observed for $m/z = 74$ in the low dose irradiated $CO-CH_3CH_2OH$ ice (Fig. 4e). We have added the following sentence in the Supplementary Information (Page S2):

“For the CO–CH₃CH₂OH system, the ion signal at $m/z = 74$ can belong to organic compounds with formulae including C₆H₂, C₄H₁₀O, C₃H₆O₂, and C₂H₂O₃. Recall that the fully isotopic labeling experiments with ¹³C and D, and the partially ¹⁸O isotopic labeled experiment confirm the inclusion of exactly six hydrogen atoms, exactly three carbon atoms, and at least one oxygen atom, hence verifying the molecular formula C₃H₆O₂ in CO–CH₃CH₂OH ice. The formulae such as C₃H₆S, C₂H₆N₂O, and CH₆N₄ are not possible.”

Methods/Experimental:

As this work offers an identification of species at m/z 74 (aldehyde containing six hydrogen), this is utmost important to talk about the contamination/decontamination of the experiment. I could not find that authors discussed this aspect. For example, there could be contaminant from previous usage of the system/apparatus or contaminant within sample itself that could result in such species that produce a signal at m/z 74.

Reply: The previous system using the apparatus was experiments studied carbon monoxide–water ices. For each experiment with electron irradiation, the chambers were vented and the sample substrate was replaced with a new polished silver wafer. Our experiments were performed at ultrahigh vacuum pressures of about 5×10^{-11} Torr. We have measured the mass spectrum of the background gases in the main chamber at 11.10 eV, at which most complex organic molecules can be ionized, showing a very clean mass spectrum (Supplementary Fig. 14a). Additionally, ethanol, ethanol-d₆, or ethanol-¹³C₂ vapor was leaked into the main chamber, and their gas phase mass spectra were recorded at 11.10 eV; no contamination molecules were detected (Supplementary Fig. 14b-14d) in these measurements. Moreover, our FTIR measurements with ice mixtures did not find IR absorptions that can be assigned to potential contamination molecules. We have added the sentence in the Method section (Pages 14):

“The gas phase mass spectra were collected for background gases in the main chamber and ethanol samples at 11.10 eV; no contamination molecules were detected (Supplementary Fig. 14).”

The same is the situation with the “D” containing species that result signal at m/z 80 in this case. For example certain species with mol. weight 74 could also give the signal that authors observed here: C₄H₁₀O, C₃H₁₀N₂, C₃H₁₀Si, C₃H₃Cl, C₂H₂O₃. Similarly, for m/z 80, one could have C₆H₈, C₄H₄N₂, C₃H₆F₂, C₂H₅ClO, C₂H₅ClO, C₂H₂CIF, CH₄O₂S, HBr. Therefore, I suggest

to address this issue as well in a substantial way. An addition of a table would be good in the Supplementary Material for better understand and the confidence on the findings.

Reply: We have added Supplementary Table 19 and the sentence in the Supplementary Information (Page S2):

“Furthermore, other potential formulae can be ruled out (Supplementary Table 19).”

Lastly, we would like to point out that different compounds likely have their own/unique temperature-programmed desorption (TPD) profile due to their desorption energies from the surface and the ice matrix. In the fully isotopic labeling experiments with ^{13}C and D , and the partially ^{18}O isotopic labeled experiment, their corresponding TPD profiles match well with that of non-isotopically labeled ice. These findings indicate the inclusion of exactly six hydrogen atoms and three carbon atoms, and at least one oxygen atom for the formula, providing convincing evidence for the assignment of the formula $\text{C}_3\text{H}_6\text{O}_2$.

Response to Reviewers

Reviewer #1 (Remarks to the Author):

My main objection to this manuscript has been the implied idea that somehow the experiments, which are of high quality in terms of physical chemistry, are simulating interstellar conditions. They do not. The authors have subsequently explained that indeed they do not represent true interstellar conditions, and the experiments have been performed to gain insight into fundamental reaction processes that may be applicable to interstellar chemistry and other venues. They have added some qualifying comments to the text in this regard. I therefore recommend publication, with one caveat. The term "Origins of Life" appears often in the text and in capital letters and italics. Although this work is distantly related to the origin of life, its repeated emphasis seems inappropriate and should be less quoted in the text.

Reply: Thank you for your suggestion. We have revised the "*Origins of Life*" to "origins of life" throughout the manuscript. Additionally, we have removed "critically linked to the *Origins of Life*" on Page 3 and "that are necessary for the *Origins of Life*" on Page 11 in the revised manuscript.

Reviewer #2 (Remarks to the Author):

The authors have addressed all the points we raised and have made appropriate additions to the manuscript

Reply: Thank you.

Reviewer #3 (Remarks to the Author):

Reply: Thank you.

Reviewer #4 (Remarks to the Author):

The authors addressed my concerns adequately.

Reply: Thank you.

Reviewer #6 (Remarks to the Author):

The authors have reasonably addressed my comments and adapted suggestions.

Reply: Thank you.